# Curcumin against Prostate Cancer: Current Evidence

**DOI:** 10.3390/biom10111536

**Published:** 2020-11-10

**Authors:** Deborah Termini, Danja J. Den Hartogh, Alina Jaglanian, Evangelia Tsiani

**Affiliations:** 1Department of Health Sciences, Brock University, St. Catharines, ON L2S 3A1, Canada; dt14fl@brocku.ca (D.T.); dd11qv@brocku.ca (D.J.D.H.); aj11fo@brocku.ca (A.J.); 2Centre for Bone and Muscle Health, Brock University, St. Catharines, ON L2S 3A1, Canada

**Keywords:** prostate cancer, proliferation, apoptosis, survival, androgen-sensitive, androgen-insensitive, polyphenol, curcumin, in vitro, in vivo

## Abstract

Cancer is a condition characterized by remarkably enhanced rates of cell proliferation paired with evasion of cell death. These deregulated cellular processes take place following genetic mutations leading to the activation of oncogenes, the loss of tumor suppressor genes, and the disruption of key signaling pathways that control and promote homeostasis. Plant extracts and plant-derived compounds have historically been utilized as medicinal remedies in different cultures due to their anti-inflammatory, antioxidant, and antimicrobial properties. Many chemotherapeutic agents used in the treatment of cancer are derived from plants, and the scientific interest in discovering plant-derived chemicals with anticancer potential continues today. Curcumin, a turmeric-derived polyphenol, has been reported to possess antiproliferative and proapoptotic properties. In the present review, we summarize all the in vitro and in vivo studies examining the effects of curcumin in prostate cancer.

## 1. Introduction

### 1.1. Prostate Cancer

Cancer is a disease caused by genetic and environmental factors. It is characterized by the accumulation of DNA mutations that activate genes coding for proteins driving proliferation (proto-oncogenes) or inhibit genes that induce apoptosis and tumor suppression (tumor suppressor genes) [1]. Alterations in cellular signaling mechanisms lead to the development of hallmark capabilities of cancer cells such as genome instability/mutation, enabling replicative immortality, dysregulating cellular energetics, uncontrolled proliferation, evasion of growth suppressors, resisting apoptosis, tumor-promoting inflammation, inducing angiogenesis, metastasis to different body tissues away from the tumor’s original location, and avoiding the immune system [2,3]. Prostatic neoplasm is characterized as a carcinoma initiating in or affecting the prostate epithelium in men.

Prostate cancer is the second most diagnosed type of cancer and the fifth leading cause of death in the worldwide male population, with mortality rates increasing from 150,000 to 250,000 between 1990 and 2010. In 2018, 1,276,106 new cases were diagnosed and 358,989 deaths were reported [4,5]. The epidemiology of the disease varies amongst different countries; it is most commonly present in European, Australian, North American, and African American men, whereas, in Asia, prostate cancer has a lower prevalence [6]. The risk factors for prostate cancer include family history, obesity, old age, and ethnicity [5].

Prostate cancer is classified as either androgen-sensitive or -insensitive, indicative of its ability to respond to stimulation by testosterone (Figure 1). Androgens normally promote growth and survival of the prostate epithelium by binding and activating the androgen receptor (AR) [7,8,9]. Upon intranuclear compartmentalization from the cytoplasm and DNA binding, the AR–androgen complex acts as a nuclear transcription factor for the activation of genes promoting the synthesis of prostate-specific antigen (PSA) and proteins involved in cell proliferation [9]. Early-stage prostate cancer is heavily dependent on AR activation for survival, but reoccurrence is in most cases, characterized by androgen-independent tumors due to adaptations to low androgen levels [9]. Surrogate AR pathways originate when signaling is amplified due to increased receptor sensitivity or may occur without the need for androgen binding altogether [9,10]. Prostate cancers with a loss of AR function bypass androgen receptor signaling and activate different survival pathways to promote metastasis [9]. 

Generally, androgen-insensitive carcinomas represent later-stage tumors, which are more aggressive and have high metastatic potential, whereas androgen-sensitive prostate cancer is highly responsive to and is likely controlled with androgen deprivation therapy [11,12].

The main treatments that are available for prostate cancer include radiation therapy, surgery, chemotherapy, and hormonal therapy (Figure 1). The treatment delivered to each patient depends on the nature of the tumor and the probability of success and reoccurrence while trying to reduce adverse events. Surgery is most often performed in high-risk cases with advanced in situ carcinoma [13], and, although the reduction in metastatic progression, as well as localized tumor progression, is remarkably reduced, it is associated with a relatively low reduction in mortality risk after 10 years [14].

Another method for high-risk localized tumors or in patients with localized tumors and higher life expectancy is radiotherapy, specifically, external-beam radiotherapy and brachytherapy [15]. Low-dose brachytherapy methods generally involve the long-term insertion of radioactive seeds with a half-life of approximately 60 days, whereas high-dose rate brachytherapy is carried through exposure to higher doses of radiation in a relatively shorter time span [16].

Although AR signaling is important for normal prostate tissue homeostasis, its overactivation is well established to be involved in prostate cancer formation [10,17]. For this reason, androgen deprivation is regarded as one of the most aggressive and successful initial treatments for malignant prostate cancer [18]. Androgen deprivation therapy can also be used in combination with surgery and radiation therapy, and it is often performed in palliative care patients as well [19]. In order to achieve androgen deprivation, medications such as luteinizing hormone-releasing hormone, estrogen, gonadotropin-releasing hormone antagonists, AR blockers, and other inhibitor of steroid synthesis, are used [11]. Some controversy is attached to androgen deprivation therapy as it seems to be associated with adverse events including, but not limited to, erectile dysfunction, hot flashes, anemia, and depression [18]. Chemotherapy is often not considerably effective against prostate cancer progression, although it has been shown to provide successful results in patients with androgen insensitivity [20]. Common drugs for chemotherapeutic treatment are docetaxel, paclitaxel, mitoxantrone, and doxorubicin, which are often used in combination to produce greater effects [21]. Table 1 summarizes the common treatments for localized and metastasized prostate cancer.

The understanding of prostate cancer cell biology has been significantly advanced by the utilization, for research purposes, of prostate cancer cell lines. The use of prostate cancer cell lines and the establishment of in vitro prostate cancer models are paramount to the exploration of the anticancer effects of novel compounds. There are many advantages of using cell cultures as the cellular environment can be easily manipulated to examine different conditions. Importantly, in vitro studies using human cancer cells provide the opportunity to examine toxicity and establish concentrations that are effective at the cellular level. In addition, the cellular signaling mechanism could be investigated by performing experiments that overexpress or eliminate the expression of a specific protein and finding potential drug targets [29,30,31]. This critical and in-depth information is unobtainable through the study of whole organisms. The most commonly utilized cell lines in prostate cancer research, and their representative prostate cancer subtypes are described in Table 2.

Surgery and radiation therapy are inefficient against metastasized tumors, hormone deprivation therapy is ineffective against androgen-insensitive/castrate-resistant cancers, and chemotherapy’s adverse effects may heavily impair a patient’s quality of life. The need for new therapeutic agents is strongly required in the treatment of prostate cancer. Use of novel compounds that target the specific signaling cascades associated with enhanced survival and metastatic potential of prostate cancer will increase patient survival rate, while avoiding the detrimental effects associated with chemotherapy.

### 1.2. Curcumin

Many pharmaceutical drugs have been developed by researching and analyzing compounds derived from plants. These include aspirin, which contains salicylic acid derived from willow bark [49], morphine derived from *Papaveraceae somniferum* (opium poppy) [50], and chemotherapeutic drugs including paclitaxel (taxol) derived from *Taxus brevifolia* (Pacific Yew) [51], vinblastine and vincristine derived from the Madagascar periwinkle plant (*Catharanthus roseus*), taxotere (docetaxel) derived from the European yew (*Taxus baccata*) and fungal metabolites, etoposide derived from the roots of mayapple plants (*Podophyllum peltatam*), tenioposide derived from the wild mandrake (*Podophyllum peltatum*), and many others [52].

Turmeric derived from the *Curcuma longa* plant contains the polyphenols curcumin, demethoxycurcumin, and bisdemethoxycurcumin (Figure 2), and it has caught the attention of researchers due to its extensive use as a culinary ingredient (the bright yellow color of curry is attributed to turmeric) in most Asian countries and the many reports of its antioxidant, antimicrobial, and anti-inflammatory properties [53]. Curcumin (diferulolylmethane) is extensively utilized in a variety of settings including cosmetic and herbal supplementation, and, although its medicinal properties have been investigated for more than 30 years, its mechanisms of action and exact molecular targets remain unclear.

Many studies have examined the effects of curcumin treatment on different prostate cancer cells. These in vitro studies provide the opportunity to investigate and elucidate detailed cellular mechanisms involved in the action of curcumin that may explain its therapeutic properties. The first section of the present article summarizes the evidence provided by these in vitro studies. Combination treatments and studies utilizing curcumin as a positive control were excluded. The second section of the present article summarizes the evidence provided by in vivo studies. The studies are arranged chronologically to emphasize research progression throughout the years, and tables summarizing the cell line/animal model used, the concentration/dose of curcumin, duration of treatment, and the major findings are included to straightforwardly extrapolate important information from each study.

## 2. Effects of Curcumin on Prostate Cancer Cells In Vitro

### 2.1. Androgen-Sensitive Prostate Cancer Cells

A number of studies have examined the anticancer effects of curcumin utilizing androgen-sensitive prostate cancer cell lines (Table 3) and are summarized in Figure 3. In a study by Dorai et al., treatment with curcumin reduced the proliferation rate of LNCaP cells to 20–30% the rate observed in untreated cells, establishing curcumin’s half-maximum inhibitory concentration IC_50_ at 10–20 μM [54]. The levels of anti-apoptotic proteins B-cell lymphoma-2 (Bcl-2) and B-cell lymphoma-extra large (Bcl-xL) were remarkably suppressed, whereas the levels of Bcl-2-assocaited X (Bax) protein remained unaltered under the same conditions, indicating a higher Bax/Bcl-2 ratio compared to untreated cells. Furthermore, curcumin induced the translocation of phosphatidylserine to the outer plasma membrane and promoted the loss of structural integrity within the membrane itself, indicative of programmed cell death [54]. Comparatively, the upregulation of poly (ADP-ribose) polymerase (PARP) cleavage, associated with apoptosis progression, was further enhanced by curcumin treatment. The expression of the androgen receptor protein (AR) was significantly inhibited in curcumin-treated cells as opposed to the control, and prostate-specific antigen (PSA) levels were also decreased [54].

Treatment of LNCaP cells with curcumin resulted in significant inhibition of cell viability and induction of apoptosis [55]. Additional evidence of induction of apoptosis was provided by the increased levels of propidium iodide (PI)-stained cells. Furthermore, this study provided evidence of an important role of nuclear factor Kappa B (NF-κB) activation in cell survival. Curcumin potentiated the tumor necrosis factor (TNF)-induced apoptosis which was associated with a suppression of the TNF- induced NF-κB activation. At the cellular level, a condensation of chromatin could be observed, followed by plasma membrane blebbing and the formation of vacuoles within the protoplasm [55].

Curcumin dose-dependently inhibited LNCaP prostate cancer cell growth and reduced survival [56]. Cell co-treatment with curcumin and androgens (R1881) resulted in a remarkable reduction in AR transactivation despite androgen presence. Curcumin lowered AR transcriptional activity and AR protein expression, and it stimulated the suppression of proteins c-Jun/activator protein 1 (AP-1) and cAMP response element-binding (CREB)-binding protein (CBP) [56].

A study by Mukhopadhyay et al. also identified curcumin as an antiproliferative agent in LNCaP cells, capable of decreasing DNA synthesis efficiency in a dose-dependent manner as observed by lower thymidine incorporation [57]. Cyclin D1 protein expression, which reflects a cell’s ability to transition from the G1 to the S stage of cell division, was significantly inhibited. Cell viability was not particularly suppressed as up to 70% of cells were still deemed viable, indicating a mechanism of cyclin D1 suppression that is not correlated to curcumin’s cytotoxic effects. Additionally, cyclin D1 mRNA expression was downregulated, paired with a time-dependent inhibition of cyclin D1 promoter activity and a drop in CDK4 activity [57].

Deeb et al. reported a decrease in LNCaP cell viability following exposure to curcumin, paired with cell-cycle arrest at the G2/M phase [58]. An increased sensitivity to TNF-related apoptosis-inducing ligand (TRAIL)-mediated toxicity was also observed, whereas no significant effects on DNA fragmentation were shown when compared to the control. Similarly, the cleavage of caspases-3 and -8 was not particularly altered by curcumin treatment, although a faint reduction in the levels of proapoptotic protein Bid and mitochondrial cytochrome c was detected. The expression of procaspase-9 protein, normally activated following cytochrome c release, remained unaffected upon curcumin treatment [58].

In a study by Chaudhary and Hruska, treatment of LNCaP cells with curcumin resulted in a significant inhibition of protein kinase B (Akt) phosphorylation/activation in either serum-starved or 10% serum-containing conditions [59].

Prostate cancer cells have increased bone metastatic potential due to their osteomimetic properties. C4-2B (metastatic derivative of LNCaP) prostate cancer cells pretreated with curcumin had lower levels of ligand-mediated epidermal growth factor receptor (EGFR) autophosphorylation when compared to the control, consequently hindering the EGFR ligand from promoting downstream signaling [48]. Curcumin inhibited C4-2B cell mineralization as shown by the presence of fewer van Kossa stain-positive nodules. Curcumin interference with the expression of osteoblast-like properties in C4-2B cells was also confirmed by the downregulation of the Cbfa-1 transcription factor, responsible for the expression of proteins specific to bone tissue. Moreover, curcumin inhibited the activity of the NF-κB activator IκB kinase (IKK) and the expression of cyclooxygenase (COX-2) protein [48].

In a study by Deeb et al., although curcumin alone did not contribute to significant cell death (~10%), the addition of TRAIL greatly enhanced cytotoxicity [60]. TRAIL integration also enhanced curcumin’s effects on apoptosis, as shown by increased levels of phosphatidylserine on the outer surface of the cellular membrane, indicative of programmed cell death. Similarly, procaspase-3 cleavage was upregulated by curcumin in the presence of TRAIL. In addition, curcumin dose-dependently inhibited NF-κB, as well as IκBα phosphorylation, which leads to NF-κB activation [60].

In a study by Zhang et al., curcumin downregulated the expression of the NKX3.1 gene and NK3X3.1 protein in LNCaP cells in a dose-dependent manner [69]. Transfection with a vector containing a NKX3.1 promoter fragment and treatment of transfected cells with curcumin resulted in an inhibited promoter activity as well. Curcumin inhibited R1881 androgen-induced NKX3.1 protein expression, and it suppressed AR gene and protein expression [69].

Another article published by Li et al. indicated an inhibition in the expression of the mouse double minute 2 homolog (MDM2) oncogene in LNCaP cells in a dose-dependent manner following treatment with curcumin [70]. A positive relationship between MDM2 inhibition and the inhibition of the phosphoinositide 3-kinase (PI3K)/mechanistic target of rapamycin (mTOR)/E-twenty six proto-oncogene 2 (ETS2) pathway was identified in PC3 prostate cancer cells [70].

Shankar and Srivastava found an inhibition in LNCaP cell viability and colony formation efficiency following treatment with curcumin [61]. Caspase-3 activation and nuclear translocation, indicative of apoptosis, was upregulated, paired with an increase in PARP cleavage. Bcl-2 protein expression was inhibited whereas Bax and Bak proteins were upregulated following treatment with curcumin. Additionally, the expression of proapoptotic genes Bim, Bax, Bak, p53 upregulated modulator of apoptosis (PUMA), and Noxa was augmented, whereas the expression of Bcl-2 and Bcl-xL genes was reduced [61]. Curcumin stimulated a drop in the mitochondrial membrane potential, and it promoted the release of mitochondrial proteins Smac/DIABLO, cytochrome c, and Omi/HtrA2 to the cell’s cytoplasm. Comparably, the translocation of Bax and p53 to the mitochondria was also increased. The phosphorylation and acetylation of p53 was also increased, although protein levels remained the same, suggesting a post-translational mechanism of action of curcumin in the modulation of tumor suppressor genes. Curcumin promoted the production of reactive oxygen species (ROS), and it inhibited the expression of p110 and p85 subunits of PI3K in a dose-dependent fashion while reducing Akt phosphorylation, suggesting potent inhibitory effects of curcumin on the PI3K/Akt pathway. Furthermore, use of inhibitors or dominant negative Akt to downregulate Akt enhanced the curcumin-induced apoptosis, while the use of constitutively active Akt attenuated apoptosis. The inhibition of Akt phosphorylation/activation was positively associated with the downregulation of curcumin-induced release of Smac from mitochondria and p53 translocation to mitochondria, indicating its role in the suppression of apoptosis [61].

In a study by Deeb et al., phosphorylated Akt levels in LNCaP cells were dose-dependently inhibited by curcumin without an effect on phosphatase and tensin homolog (PTEN) [71]. Treatment of LNCaP cells with curcumin upregulated the mitogen-activated protein kinase phosphatase-5 (MKP5) and reduced cytokine-induced inflammation, suggesting the potential of curcumin to be utilized as a prostatic anti-inflammatory agent [72].

Curcumin inhibited viability of LNCaP cells while sensitizing them to TRAIL [62]. Curcumin additionally inhibited colony formation efficiency while upregulating TRAIL-induced apoptotic processes. An upregulation of the expression of cell-surface death receptors DR4 and DR5 could also be observed, but no effects on the expression of DcR1 and DcR2 were shown. Furthermore, curcumin induced the expression of the proapoptotic proteins Bak, Bax, PUMA, Bim, and Noxa and reduced the expression of antiapoptotic proteins Bcl-xL and Bcl-2 [62]. Curcumin promoted a decrease in the mitochondrial membrane potential and enhanced the cleavage of caspases-3, -8, and -9 and PARP. All these effects were enhanced when curcumin was combined with TRAIL [62].

Treatment of LNCaP cells with curcumin induced cell-cycle arrest at the G1/S phase and promoted apoptosis in a dose-dependent fashion [63]. Curcumin enhanced the expression of the cyclin-dependent kinase (CDK) inhibitors p16, p21, and p27, while it reduced the expression of G1-phase proteins cyclin D1 and CYCLIN E. In addition, curcumin reduced the phosphorylation of the retinoblastoma (Rb) protein. Importantly, the use of siRNA to downregulate p21 abolished the curcumin-induced apoptosis. It was shown that curcumin-induced inhibition of cyclin D1 and cyclin E occurred through ubiquitin-dependent proteasomal degradation as inhibition of the 26S proteasome by lactacystin abolished the effects of curcumin on cyclin downregulation [63].

Treatment of LNCaP and C4-2B cells with curcumin inhibited cell proliferation in a dose- and time-dependent manner [73]. Additionally, treatment of both cell lines with curcumin upregulated genes whose activation leads to increased functions in methionine tRNA synthase, hemeoxygenase decyclizing activity, transcription corepressor activity, and arrest of cell growth, whereas genes stimulating kallikrein 2 and 3, androgen-regulated neural precursor cell expressed developmentally down-regulated protein 4 (NEDD4)-binding protein, transmembrane proteases, and cyclin b1 were downregulated. Curcumin also downregulated the expression of the AR protein and NKX3.1 tumor suppressor, as well as PSA levels, Erb-B2 Receptor Tyrosine Kinase 2 (ERBB2), and epidermal growth factor receptor (EGFR). PSA secretion in the media was also decreased when compared to the control [73].

Curcumin inhibited LNCaP cell proliferation, as well as PSA gene and protein expression, while reducing the basal and the interleukin 6 (IL-6)-induced AR levels [74]. LNCaP cell viability was significantly reduced, paired with increased apoptosis, when compared to the untreated group [75].

Treatment of TRAMP-C2 mouse prostate cancer cells with curcumin inhibited cell proliferation (IC_50_ = 20 μM) and downregulated the hedgehog signaling pathway. The glioma-associated oncogene (Gli1) protein levels were also significantly reduced with curcumin treatment [76].

Exposure of LNCaP cells to curcumin resulted in a remarkable induction of apoptosis that was associated with significantly increased levels of ceramide, a known mediator of cytotoxic stimulus-induced apoptosis [66]. Treatment of LNCaP cells with curcumin resulted in significant inhibition of proliferation and reduced AR, β-catenin, cyclin D1, and c-myc expression. Additionally, phosphorylation of Akt and glycogen synthase kinase-3β was attenuated, but phosphorylated β-catenin was increased [64].

Curcumin caused cytotoxicity in LNCaP and 22RV1 cells, and it appeared to induce apoptosis through activation of autophagy as demonstrated by an upregulation of the microtubule-associated protein 1A/1B-light chain 3 (LC3b-II isoform), indicative of the quantity of autophagosomes present [77]. Cell-cycle arrest at the G2/M phase occurred in 22RV1 cells. Inhibition of the expression of cyclin B1 and the proliferating cell nuclear antigen (PCNA), proteins associated with cell-cycle progression, was observed. Transcriptional activity was heavily decreased in 22RV1 cells shown via the indirect inhibition of β-catenin, potentially caused by a decreased expression of transcription factor 4 (TCF-4) and the co-repressors CBP and p300, associated with the transcriptional activity of Wnt within the Wnt/β-catenin/TCF-4 pathway. In 22Rv1 cells, the expression of β-catenin target genes c-myc, surviving, and cyclin D1 was inhibited [77].

Treatment of LNCaP cells with curcumin blocked the hypoxia-induced expression of PSA. This effect seemed to be mediated by a significant reduction in AR protein levels without altering the levels of either hypoxia-inducible factor 1-alpha (HIF-1α) or vascular endothelial growth factor (VEGF) protein levels [78].

The normal hypermethylation of the first five CpG in the CpG island of the nuclear factor erythroid 2–related factor 2 (Nrf2) gene in TRAMP-C1 prostate cancer cells was reversed upon curcumin treatment, thus favoring the expression of the Nrf2 protein, which regulates antioxidative stress proteins, including nicotinamide adenine dinucleotide phosphate (NADPH) and glutathione peroxidases, and whose deficiency is correlated with an increased risk of tumorigenesis in mice following exposure to carcinogens [79]. Curcumin additionally stimulated the expression of the NAD(P)H quinone dehydrogenase 1 (NQO-1) gene, a downstream target of Nrf2, and it inhibited the rate of activity of a DNA methyltransferase analogue, M.SssI, as shown by the presence of numerous unmethylated sections following curcumin treatment when compared to the control. The expression of DNA methyltransferases DNMT 1, 3A, and 3B remained relatively unchanged [79].

Treatment of LNCaP cells with curcumin induced both cytotoxicity and a change in the methylation status of LNCaP cells [80]. An inhibition of trimethylation of histone 3 Lys 27 (H3K27me3) was observed, paired with decreased methylation of the CpG sites in the Neurog1 gene (by ~20%) and binding of methylated CpG-binding protein (MECP2) to Neurog1, leading to a decreased methylation of the Neurog1 promoter region. Additionally, an increase in total levels of Neurog1 mRNA was shown, leading to increased Neurog1 protein levels. Curcumin had no effect on the expression of DNA methyl-binding proteins MBD2 and MeCP2, and it did not impact the levels of DNA methyltransferase (DNMT1 and DNMT3A). Despite promoting an overall inhibitory effect on histone deacetylase activity, curcumin favored the expression of histone deacetylase (HDAC)1, HDAC4, HDAC5, and HDAC8, whereas only a minor increase (~20%) in HDAC3 mRNA expression was observed [80].

The 22RV1 cells treated with curcumin differentially expressed 32 proteins when compared to the untreated group, including proteins known to be involved in drug resistance such as moesin (MSN) and RNA binding motif protein 17 (RBM17). Proteins involved in cell death such as high mobility group box 1 protein (HMGB1) and nucleophosmin 1 (NPM1), as well as protein phosphatase 2 (PP2R1A) and members of the heat-protein (HSP) family members HSP90B1, HSP90AB1, and HSPA9 were modulated [81]. In 22RV1 cells, an enhancement in the expression of HMGB1, HSP90AA1, gap junction alpha-1 (GJA1), protein kinase C epsilon (PRKCE), x-ray repair cross-complementing protein 6 (XRCC6), MIR-141, and MIR-183 was observed, and, in both LNCaP and 22RV1 cells, curcumin decreased the expression of the androgen receptor (AR) [81].

Curcumin inhibited cell proliferation in LNCaP and C4-2B cells while attenuating the nuclear signaling of β-catenin through phosphoinositide-dependent kinase 1 (PDK1) upregulation, which favored β-catenin compartmentalization within the membrane and the nucleus [82]. In C4-2B cells, curcumin decreased the overall expression of nuclear β-catenin and inhibited its transcriptional activity, leading to the potential inhibition of molecules which possess oncogenic properties, including c-myc, c-jun, and cyclin D. Curcumin further enhanced the expression of inactive phospho-cofilin, a downstream target of PDK1. Cell–cell aggregation was also improved upon treatment, possibly due to greater localization of β-catenin within the cell membrane [82].

Invasion/migration and matriptase activation in LNCaP cells stimulated by exposure to androgens (DHT) were inhibited by curcumin [83]. EGF-induced expression of matriptase was also downregulated by curcumin treatment. Total and activated matriptase was inhibited in curcumin treated CW22RV1 and C-33 cells, whereas matripase gene expression remained unaltered indicating involvement of a post-translational mechanism. No change in the expression of total and activated matriptase was detected in PNT-2 normal prostate epithelial cells [83].

Curcumin elicited an increase in programmed cell death in LNCaP cells [84]. Bcl-2 protein expression was inhibited, whereas Bax was upregulated. Curcumin also induced a significant production of cytosolic and mitochondrial O_2_^−^, as well as the release of H_2_O_2_ in the extracellular media and the expression of copper-zinc-superoxide dismutase (CuZnSOD), processes indicative of increased ROS synthesis. Curcumin inhibited cell viability following pretreatment with N-acetyl cysteine (NAC), an antioxidant precursor of glutathione (GSH), and it increased the phosphorylation of apoptosis signal-regulated kinase 1 (ASK1), a protein associated with the initiation of apoptotic processes, leading to cytotoxicity. Furthermore, curcumin enhanced the levels of thioredoxin 1 (TRX1) mRNA and protein, increased its oxidation levels, and increased its nuclear translocation [83].

A dose-dependent inhibition of LNCaP and 22RV1 cell proliferation was observed following treatment with curcumin nanoparticles [85]. The activities of caspases-3 and -7 were increased, indicative of apoptosis. In addition, curcumin reduced testosterone production by the cells by decreasing the expression of steroidogenic proteins. Steroidogenic acute regulatory (StAR) and cytochrome P450 family 11 subfamily A member 1 (CYP11A1), proteins contributing to the conversion of cholesterol into androgens and pregnenolone, respectively, were suppressed upon curcumin treatment [85].

Treatment of LNCaP cells with curcumin induced apoptosis paired with lower phosphorylation of c-Jun NH2-terminal kinase (JNK) and c-Jun and lower levels of BCL-2 mRNA [86]. At the epigenetic level, curcumin stimulated the reduction of trimethylation of H3K4 (H3K4me3) leading to repressed gene expression, although no significant changes were observed in H3K27 gene expression [86].

LNCaP and C4-2B cells treated with curcumin had reduced Myc protein levels, and increased heme oxygenase-1 (HMOX1), Ewing sarcoma breakpoint region 1 (EWSR1), cyclic AMP-dependent transcription factor (ATF3), sestrin-2 (SESN2), ferritin heavy chain 1 (FTH1), glutamate–cysteine ligase modifier subunit (GCLM), AF4/FMR2 family member 4 (AFF4), E74-like ETS transcription factor 3 (ELF3), Ras like without CAAX 1 (RIT1), DRE1, DNA damage inducible transcript 3 (DDIT3), cytoplasmic polyadenylation element-binding protein 4 (CPEB4), cAMP-responsive element modulator (CREM), baculoviral IAP repeat-containing protein 4 (BIRC4), RIO kinase 3 (RIOK3), phosphoinositide 3-kinase regulatory subunit 3 (PIK3R3), ubiquitin-conjugating enzyme E2 H (UBE2H), and chromosome 6 open reading frame 62 (C6orf62) mRNA levels [87]. Curcumin inhibited RAF1, BCL6, and insulin like growth factor 1 receptor (IGF1R) mRNA levels and increased PTEN, epidermal growth factor receptor 1 (EGFR1), SMAD, forkhead box protein O3 (FOXO3), AKT1, and RAD51 mRNA levels in LNCaP cells. In addition, PTEN-stimulated cell-cycle arrest, apoptosis, and cell–cell adhesion were increased, while transforming growth factor beta (TGF-β), WNT, AP-1, NF-κB, and PI3K/Akt/mTOR signaling pathways were reduced with curcumin. However, in C4-2B cells, SOX4, EGFR, Wilms tumor 1 (WT1), and E2F transcription factor 2 (E2F2) mRNA levels were downregulated, and metastasis-associated lung adenocarcinoma transcript 1 (MALAT1) mRNA level was enhanced with curcumin treatment. On the other hand, in C4-2B cells, curcumin enhanced IL-6 signaling, PTEN-dependent cell-cycle arrest, and apoptosis, whereas TGF-β and WNT signaling pathways were suppressed [87].

### 2.2. Androgen-Insensitive Prostate Cancer Cells

Many studies have examined the anticancer effects of curcumin also utilizing androgen-insensitive prostate cancer cell lines (Table 4) and are summarized in Figure 3. A study by Dorai et al. found a 60–80% inhibition of PC3 prostate cancer cell proliferation by curcumin treatment [54].

Treatment of DU-145 prostate cancer cells with curcumin resulted in reduced proliferation and increased apoptosis paired with a downregulation in the expression of NF-κB [55]. Curcumin further inhibited the expression of nuclear transcription factor activator protein-1 (AP-1), composed of c-JUN and c-Fos. Additionally, the expression of the antiapoptotic proteins Bcl-2 and Bcl-xl was inhibited, whereas the expression and enzymatic activity of the proapoptotic proteins procaspase-3 and procaspase-8 were increased following curcumin treatment [55].

Curcumin dose-dependently inhibited cell growth and colony-forming capabilities of PC3 cells. Furthermore, curcumin promoted lower AR transcriptional activity and the suppression of proteins c-Jun/AP-1 and cAMP response element-binding protein-binding protein (CBP), while also downregulating the transcription of NF-κB [56].

Deeb et al. observed fewer viable DU-145 cells paired with cell-cycle arrest at the G2/M phase following exposure to curcumin [58].

LNCaP and PC-3 prostate cancer cells were found to have constitutively activated Akt, and treatment with curcumin completely abolished Akt activation [59]. However, curcumin did not block the serum-induced Akt activation in DU-145 cells [59]. These data indicate different effects of curcumin on Akt activation in different prostate cancer cells, and the authors hypothesized that the anticancer potential of curcumin may be due to its ability to inhibit/target Akt. A study by Hong et al. showed a dose-dependent inhibition of DU-145 cell proliferation [88]. Curcumin treatment enhanced apoptotic activity and reduced matrix metallopeptidase protein (MMP)-2 and MMP-9 secretion [88].

Curcumin induced apoptosis and decreased viability and proliferation of PC3 cells [70]. Curcumin independent of p53 inhibited the expression of MDM2 in PC3 cells while inducing the expression of p21 and Bax and suppressing E2F1 and Bcl-2 [70]. Additionally, the transcription factor erthroblastosis virus transcription factor 2 (ETS2), whose overexpression is correlated with enhanced transcription of MDM2, was inhibited by curcumin [70]. Knockdown of ETS2 abolished curcumin’s effects on the expression of MDM2. Exposure to curcumin inhibited the PI3K/Akt/mTOR pathway [70]. These data suggest that the anticancer effects of curcumin are due to its ability to downregulate the MDM2 oncogene by inhibiting the PI3K/Akt/mTOR pathway.

Curcumin reduced viability of PC3 and DU-145 prostate cancer cells without affecting normal prostate epithelial cells [60]. The colony formation efficiency of either prostate cancer cell line was decreased, while caspase-3 activation and its nuclear translocation were increased by curcumin [61].

Treatment with curcumin did not affect viability of DU-145 and PC3 cells, although the phosphorylation and degradation of the NF-κB inhibitor IκBα was reduced only in DU-145 cells [71]. Curcumin inhibited phosphorylated Akt and NF-κB protein levels in DU-145 cells, while overexpression of Akt1 prevented curcumin’s inhibition of NF-κB. Furthermore, in PC3 cells only, curcumin inhibited antiapoptotic NF-κB-dependent Bcl-2, Bcl-xL, and X-linked inhibitor of apoptosis protein (XIAP) protein levels [71].

In a study by Nonn et al., treatment of DU-145 and PC3 cells with curcumin increased the mitogen-activated protein kinase phosphatase-5 (MKP5) mRNA levels [72]. In DU-145 cells, pretreatment with curcumin blocked the TNFα- and IL-1β-induced p38 phosphorylation and cyclooxygenase-2 (COX-2) mRNA and protein levels. Curcumin also reduced IL-6 and IL-8 mRNA levels. Additionally, a reduction of p38-mediated proinflammatory signaling by TNFα and IL-1β was observed with curcumin treatment [72].

Shankar et al. found that treatment of PC3 cells with curcumin reduced viability and induced apoptosis [62]. Curcumin also enhanced the TRAIL-mediated apoptosis [62]. The expression of cell-surface death receptors DR4 and DR5 was enhanced, while DcR1 and DcR2 expression was not affected by curcumin treatment. Furthermore, curcumin treatment increased the levels of proapoptotic proteins Bak, Bax, PΜMA, Bim, and Noxa, promoted Bid cleavage to tBid, and reduced the levels of the antiapoptotic proteins Bcl-xL and Bcl-2, as well as the levels of the inhibitors of proapoptotic proteins and XIAP [62]. Curcumin caused a decrease in the mitochondrial membrane potential and upregulated caspase-3 activity [62].

Treatment of PC3 cells with curcumin resulted in cell-cycle arrest at the G1/S phase and induction of apoptosis [63]. The levels of p27, p21, and p16 were increased, while the protein levels of cyclin D1, cyclin E, and CDK4 were reduced by curcumin treatment. Curcumin suppressed hyperphosphorylation of Rb without affecting Rb protein levels [63].

Curcumin suppressed PC3 cell proliferation by inhibiting glyoxalase 1 activity. Additionally, curcumin led to cytotoxic effects and the initiation of necrosis [89].

Treatment of PC3 cells with curcumin resulted in reduced proliferation associated with reduced levels of phosphorylated Akt and mTOR [65]. Furthermore, levels of p70S6K, forkhead box protein O1 (FoxO1), S6, and glycogen synthase kinase 3β (GSK3β) were also reduced by curcumin treatment. On the other hand, curcumin was able to induce the phosphorylation and activation of AMP-activated protein kinase (AMPKα) and its substrate acetyl-CoA carboxylase (ACC), whereas mitogen-activated protein kinase (MAPK) expression was enhanced, including extracellular signal-regulated kinase (ERK)1/2, JNK, and p38 MAPK. There was no change in the levels of PDK1, protein kinase C (PKC), or total protein levels of Akt, mTOR, p70S6K, S6, and 4E-BP1, while cyclin D1 levels were reduced. Pretreatment of PC3 cells with calyculin A, a serine/threonine phosphatase inhibitor, reversed the inhibitory effects of curcumin on phosphorylated Akt, mTOR, S6, 4E-BP1, and cyclin D1. These data indicate a mechanism of action of curcumin involving increased phosphatase activity, which leads to decreased phosphorylation of proteins within the PI3K/Akt pathway and, ultimately, decreased survival and proliferation [65].

Treatment of PC3 cells with curcumin reduced the C-C motif chemokine ligand 2 (CCL2)-mediated motility and invasion. Curcumin treatment reduced PKC-stimulated cell adhesion and significantly reduced motility, adhesion, and invasion in PC3 cells overexpressing PKC [90]. In addition, curcumin effectively inhibited CCL2 mRNA levels and secretion. In the presence of phorbol 12-myristate 13-acetate (PMA), a PKC activator, curcumin effectively reduced PMA-stimulated CCL2 mRNA levels. Curcumin also reduced CCL2- and PMA-stimulated MMP-9 activity. These data indicate a potential mechanism of action in which curcumin acts as an inhibitor of CCL2 through differential expression of both PKC and MMP-9 [90].

Pc-Bra1 cells treated with curcumin had reduced viability and increased apoptosis. The necrotic population in treated cells was below 10%, indicating programmed cell death [75].

PC3 cells treated with curcumin had increased apoptosis and higher levels of single-stranded DNA (ssDNA) and ceramide [66]. Additionally, curcumin increased phosphorylation and activation of JNK and p38 MAPK, while the levels of procaspases-3, -8, and -9 were reduced, indicating caspase cleavage and activation [66]. Curcumin-induced cytochrome c accumulation within the cellular cytoplasm, which is indicative of mitochondrial inner membrane destruction and apoptosis, was not altered in a statistically significant manner following inhibition of caspases. This suggests that, although caspases may be one of the targets of curcumin, they do not play a direct role in the mechanism of curcumin-induced cell death [66].

Treatment of 22rv1 and LNCaP cells with curcumin significantly reduced cell proliferation through increased cell-cycle arrest at G2 and reduced β-catenin and Tcf-4 transcriptional activity [77]. In addition, curcumin reduced the protein levels of Tcf-4, CBP, and p300, proteins involved in the Wnt transcriptional complex, and β-catenin target genes cyclin D1 and c-myc. However, these effects of curcumin did not occur in androgen-insensitive DU145 and PC3 cells, suggesting that curcumin may be more effective during early-stage prostate cancer [77].

Treatment of PC3 cells with curcumin reduced proliferation, induced cell-cycle arrest at the G2/M phase, and increased apoptosis [91]. The signaling pathways AP-1 and NF-κB were downregulated by curcumin. These data suggest that the suppression of proliferation by curcumin may occur through cell-cycle arrest and increased cell death by downregulating the NF-κB pathway and AP-1 transcription factor [91].

PC3 cells treated with curcumin differentially expressed 47 proteins compared to untreated cells, including protein phosphatase 2 (PP2R1A) and members of the heat-shock protein (HSP) family HSP90B1, HSP90AB1, and HSPA9 [81]. Additionally, proteins known to be involved in drug resistance such as moesin (MSN) and RNA binding motif protein 17 (RBM17), and proteins involved in cell death such as high mobility group box 1 protein (HMGB1) and nucleophosmin 1 (NPM1) were increased with curcumin. An upregulation in IL-6, insulin (INS), DNA damage inducible transcript 3 (DDIT3), N-myc downstream regulated 1 (NDRG1), and MIR-152 was also observed [81].

Killian et al. investigated the effects of curcumin in PC3 cells and found an indirect downregulation of inflammatory cytokines chemokine (C-X-C motif) ligand (CXCL)-1 and CXCL2. Curcumin inhibited Iκb kinase b (IKKb), a protein responsible for the phosphorylation and downregulation of the NF-κB inhibitor IκBα. [92].

In a study by Cheng et al., treatment of PC3 cells with curcumin resulted in reduced growth and inhibited migration and invasion [83]. The epidermal growth factor-induced PC3 cell invasion was reduced in the presence of curcumin. Total and activated matriptase levels were reduced by treatment of PC3 and DU-145 cells with curcumin. No changes in the levels of Snail, E-cadherin, vimentin, or β-catenin proteins were seen with curcumin treatment [80]. Importantly, total and activated matriptase levels remained unchanged in PNT-2 normal prostate epithelial cells following curcumin treatment. Curcumin did not affect the expression of the matriptase gene, indicating a possible post-translational mechanism of action [83].

In a study by Yu et al., PC3 cells treated with curcumin had inhibited cell viability and enhanced apoptotic processes via decreased inhibitor of DNA binding 1 (ID-1) mRNA and protein levels [93].

Li et al. demonstrated that curcumin could cause a decrease in the viability of PC3 and DU-145 cells [94]. Additionally, curcumin inhibited the levels of mucin 1 C-terminal domain (MUC1-C) and NF-κB subunit p65, and it induced the phosphorylation of ERK 1/2 and stress-activated protein kinase (SAPK)/JNK. Overexpression of MUC1-C antagonized the phosphorylating effects of curcumin on ERK and SAPK/JNK [94].

Cancer-associated fibroblasts (CAFs) are activated fibroblast parts of prostate stromal cells that support cancer progression and metastasis. Exposure of PC3 cells to CAF-conditioned media increased reactive oxygen species (ROS), IL-6 receptor, chemokine receptor 4 (CXCR4) levels, and epithelial-to-mesenchymal transition (EMT), and it activated the monoamine oxidase A (MAOA)/mammalian target of rapamycin (mTOR)/hypoxia-inducible factor-1α (HIF-1α) signaling pathway [93]. All these effects were significantly reduced in the presence of curcumin [95].

A study by Hu et al. found increased DU-145 cell death upon treatment with curcumin [96]. DU-145 cells were co-treated with human growth factor (HGF) to assess curcumin’s potential cytotoxic activity under conditions favorable for cancer growth. HGF-induced cell scattering and wound closure were inhibited in the presence of curcumin, indicating effects against cell invasion. The upregulation of vimentin and the inhibition of E-cadherin, two processes which promote EMT, were promoted by growth factors, and were reversed in cells co-treated with curcumin. Furthermore, the HGF-induced increase in c-Met, Snail expression, and phosphorylation of ERK were suppressed by curcumin [96]. In contrast, phosphorylated Akt levels remained unchanged in both groups. These data suggest that the anticancer effects of curcumin are due to inhibition of the c-met/ERK/snail pathway [96].

Treatment of DU-145 cells with curcumin resulted in reduced proliferation and enhanced apoptosis that was associated with inhibition of the Notch signaling pathway [97]. In cells overexpressing Notch1, curcumin effectively reduced both proliferation and survival and stimulated apoptosis, indicating a strong potential of curcumin to inhibit Notch1 signaling. Additionally, curcumin induced cell-cycle arrest at the G0/G1 stage, reduced CDK-2 and cyclin D1 levels, and increased p27 and p23 levels [97].

DU-145 and PC3 cells treated with curcumin had increased levels of apoptosis, activated caspases-3 and -9, and increased autophagy [98]. The levels of transferrin receptor 1 (TfR1) and iron regulatory protein 1 (IRP1) were increased by curcumin treatment, indicating its iron deprivation potential. The apoptosis and autophagy induced by curcumin were counteracted by ferric ammonium citrate, indicating that curcumin’s cytotoxic effects are partially dependent on its iron-chelating properties.

Another study by Yang et al. found an inhibition of DU-145 and PC3 cell viability and proliferation with curcumin treatment [99]. In DU-145 cells, wound healing was also inhibited, with no effect on the expression of Notch1 and notch intracellular domain (NICD). On the other hand, the expression of genes MT1-MMP and MMP2, associated with cell migration, was downregulated by curcumin [99].

Cao et al. treated DU-145 and PC3 cells with curcumin and found a significant upregulation of miR-143, which was associated with decreased proliferation and migration [100] Silencing this microRNA reversed curcumin’s effects. In DU-145 cells specifically, miR-143 upregulation led to a subsequent decrease in the levels of phosphoglycerate kinase 1 (PGK1), a prostatic oncogene [98]. The expression of FOXD3, a transcription factor which has potential for binding to the promoter region of miR-143, was enhanced by curcumin [100].

Treatment with curcumin reduced viability, induced cell-cycle arrest and apoptosis, and increased ROS levels in PC3 cells [67]. At the molecular level, the expression of cleaved caspase-3 was increased and, concomitantly, uncleaved caspases-3/9 and PARP were upregulated. Caspase-12, a regulator of endoplasmic reticulum (ER) stress-induced apoptosis, was enhanced upon treatment, as well as certain ER stress markers GRP78, inositol-requiring enzyme 1, and calreticulin. Phosphorylated EIF2α, another regulator of apoptosis induction, was increased by curcumin [67]. In addition, curcumin induced autophagy in PC3 cells as shown by the increased expression of LC3B [67]. Overall, these data provide evidence of activation of the PERK/eIF2α/ATF4 pathway and induction of apoptosis by increasing the ER stress by curcumin.

In a study by Rodriguez-Garcia et al., treatment of PC3 cells with curcumin led to the induction of apoptosis that was associated with increased mRNA and protein levels of the ASK1 redox regulator TRX1 [84]. ROS production and CuZnSOD levels were enhanced with curcumin treatment [84].

A study by Zhu et al. investigated the effects of curcumin on the PC3 and DU-145 cell lines [68]. Cell viability, DNA synthesis, and cyclin D1, PCNA, β-catenin, and c-myc protein levels were reduced, while the levels of p21 were increased by curcumin treatment. Furthermore, curcumin increased the levels of miR-34a, a potent tumor suppressor, and the use of a miR-34a inhibitor abolished curcumin’s effects on cell viability [68].

### 2.3. Prostate Cancer Stem Cells

Treatment of CD44^+^/CD133^+^ human prostate cancer stem cells isolated from populations of DU-145 and 22Rv1 cells with curcumin resulted in reduced cell-cycle progression (Ccnd1 and Cdk4) and stem-cell marker (Oct4, CD44, and CD133) gene expression [101]. Curcumin increased miR-145 hybridization signal and reduced 1ncRNA–ROR hybridization signal. Furthermore, curcumin increased miR-3127, miR-3178, miR-1275, miR-3198, and miR-1908 mRNA levels and reduced miR-494, miR-193b, miR-671-5p, miR-210, miR222, miR-23b, miR-664, and miR-183 mRNA levels (Table 5) [101].

A similar study, conducted by Zhang et al. a year later, investigated the effects of curcumin on stem cells extracted from populations of DU-145 and 22RV1 cells (Table 5) [102]. Curcumin treatment significantly inhibited cell proliferation and migration. In addition, curcumin increased the mRNA levels of miR-770-5p, miR-411, and miR-1247 within the delta-like homolog 1 gene (DLK1)– type III iodothyronine deiodinase gene (DIO3) miRNA cluster, whereas the mRNA levels of miR-382 and miR-654-3p were reduced [102].

## 3. Effects of Curcumin on Prostate Cancer In Vivo

Many studies have examined the effects of curcumin treatment on animals (mice) xenografted with different human prostate cancer cells and is presented in Table 6 and Figure 4.

In a study by Dorai et al., administration of curcumin to athymic nude mice heterotopically implanted with LNCaP cells resulted in significantly reduced cell proliferation and increased apoptosis as indicated by the increased in situ pycnotic brown staining nuclei [103]. Prostate tumor growth was reduced by more than 70% with curcumin treatment. Immunohistochemical examination of the tumor indicated significant necrosis characterized by decreased mitosis, reduced nuclear cytoplasm ratio, and less prominent nucleoli with curcumin administration. Furthermore, the majority of tumors exhibited typical fibrotic features. Curcumin administration inhibited LNCaP tumor angiogenesis and microvessel density, and it had negligible endothelial cell-specific CD31 expression [103].

SCID mice were implanted with DU-145 cells in the tail veins, followed by administration of curcumin via oral gavage [88]. Curcumin administration significantly reduced the mean tumor volumes, without having an effect on mouse body weight. In addition, curcumin administration increased caspase-3 activity and reduced MMP-2 and MMP-9 activity within the tumor tissue, indicating that curcumin can promote apoptosis and inhibit MMP secretion at the location of the tumor to prevent cancer progression [88].

In a study by Khor et al., male NCr immunodeficient mice were intraperitoneally injected in the right flank with PC3 cells that were treated with curcumin starting 1 day before the injection and continually injected with curcumin daily or intraperitoneally injected with curcumin after PC3 establishment. Curcumin-treated mice possessed tumors with higher levels of apoptotic cells compared to the control, vehicle-injected group [104]. Tumor growth and volume were reduced in the curcumin-treated mice group intraperitoneally injected with curcumin prior to tumor establishment. Conversely, in mice with established PC3 tumors, curcumin did not significantly alter tumor volume, although a difference in body weight compared to the control was observed. Curcumin-treated mice additionally possessed a higher percentage of apoptotic tumor cells and a lower percentage of proliferative cells, paired with increased expression of caspase-3 and PARP, as well as the suppression of phosphorylated Akt, GSK3Bα, BAD, IKKBα, and IκBα [104].

Oral administration of curcumin to PC3 tumor-bearing nude mice resulted in significantly reduced tumor growth [70]. Curcumin administration also enhanced the antitumor effects of the chemotherapy drug gemcitabine and irradiation by further reducing tumor growth. Additionally, curcumin alone or in combination with gemcitabine or irradiation reduced MDM2 protein levels, suggesting a novel mechanism that may be essential for curcumin’s anticancer effect [70].

Administration of curcumin to Balb c nude mice implanted with TRAIL-resistant LNCaP prostate cancer cells inhibited tumor growth through increased apoptosis, enhanced sensitization to TRAIL-induced apoptosis, and reduced proliferation [105]. Curcumin upregulated TRAIL-R1/DR4, TRAIL-R2/DR5, Bax, Bak, p21/WAF1, and p27/KIP1 expression, reduced cyclin D1, vascular endothelial growth factor (VEGF), uPA, MMP-3, MMP-9, Bcl-2, and Bcl-XL expression, and the activation of NF-κB in xenograft tumors. In addition, tumor angiogenesis was inhibited with reduced number of blood vessels in the tumors and circulating EGFR-2-positive endothelial cells. These data suggest that curcumin administration can increase the sensitivity of tumor cells to TRAIL and increases tumor cell apoptosis [105].

Mice xenografted with prostate adenocarcinoma (TRAMP) cells were fed the AIN-76A diet and dietary phytochemicals curcumin or phenylethylisoathiocyanate (PEITC) either alone or in combination. Supplementation with curcumin significantly reduced prostate tumor formation [106]. Immunohistochemistry showed curcumin alone or in combination with PEITC significantly inhibited high-grade prostatic intraepithelial neoplasia (PIN) expression, reduced proliferation, and increased apoptosis. In addition, curcumin supplementation increased proapoptotic Bad and cleaved caspase-3 protein levels. Furthermore, supplementation of curcumin alone or curcumin and PEITC may mechanistically decreased cell proliferation by downregulating the Akt signaling pathway as curcumin reduced phosphorylated PDK1, Akt, and FOXO1/FKHR levels [106].

In a study by Narayanan et al., co-gavage administration of liposome encapsulated curcumin and resveratrol to PTEN-knockout (PTENKO) mice resulted in significantly reduced weight gain and prostate weight compared to control C57BL6/J mice [107]. The co-administration of liposome encapsulated curcumin and resveratrol reduced epithelial cell proliferation and mPIN lesions, resulting in significantly reduced prostate adenocarcinomas. In addition, the bioavailability of curcumin and resveratrol was significantly improved by the encapsulation process, suggesting that the use of liposomes and the combined administration of curcumin and resveratrol can effectively prevent prostate cancer growth [107].

In a study by Fernandez-Martinez et al., pretreatment of PC3 cells with curcumin for 24 h before injection into nude mice resulted in significantly reduced vasoactive intestinal peptide (VIP)-induced PC3 cell proliferation and tumor growth and progression, indicating anticancer properties [71]. In addition, vasoactive intestinal peptide (VIP) and VEGF mRNA and MMP-2, MMP-9 and VEGF protein levels were significantly reduced with curcumin pre-treatment, suggesting reduced angiogenesis and metastasis. VPAC_1_ receptor density, the primary membrane receptor for VIP, was significantly reduced with curcumin pretreatment [108].

In a study by Killian et al., CD-1 Foxn1^nu^ male mice were injected with PC3 cells into the heart of the mice and subsequently fed a diet supplemented with curcumin [92]. Curcumin-treated and control animals had tumor cells present in the intrapulmonary and peripulmonary compartments. The vitality of the tumor cells was reduced with curcumin treatment, determined via proliferating Ki-67 positive cells. The tumors of curcumin-treated and control animals were similar in dimension, morphology, and histology. Additionally, curcumin treatment prevented the formation of lung metastases, and the size of the tumors was significantly reduced [92].

In normal prostate, protein kinase D1 (PKD1) is highly expressed; however, the progression of prostate cancer is associated with its decrease. Activated PKD1 phosphorylates downstream targets E-cadherin and β-catenin, leading to enhanced cell–cell adhesion, and inhibits the transcriptional activity of β-catenin and AR, resulting in reduced cell proliferation. Administration of curcumin via intratumoral injection in athymic nude mice subcutaneously inoculated with androgen-independent C4-2 cells significantly reduced tumor growth compared to control tumor-bearing mice [82]. Additionally, curcumin administration increased β-catenin subcellular localization in tumor tissues, suggesting that curcumin may inhibit prostate tumor growth by modulating PKD1/β-catenin activity [82].

In a study by Cheng et al., male nude mice inoculated subcutaneously with luciferase-expressed PC3 cells and injected intraperitoneal daily with curcumin had significantly reduced prostate cancer tumor growth and metastasis. In addition, curcumin treatment reduced the number of metastatic lesions at the brachial lymph nodes and the level of activated matriptase in the xenografted mice [83].

In a study by Yallapu et al., athymic nude mice, subcutaneously injected with C4-2 cells, were intratumorally injected with curcumin or novel poly(lactic-*co*-glycolic acid)–curcumin nanoparticles (PLGA–CUR NPs), resulting in significantly reduced tumor volume compared to controls, DMSO, or PLGA-NPs [109]. The antitumor effect of PLGA–CUR NPs was greater than curcumin, suggesting that the addition of nanoparticles may improve the efficacy of curcumin. Immunohistochemical analysis showed that curcumin and PLGA–CUR NP treatment significantly reduced Bcl-xL and nuclear AR levels. In addition, curcumin and PLGA–CUR NP treatment increased the membrane staining of β-catenin and reduced the nuclear β-catenin activity [109]. Treatment with PLGA–CUR NPs significantly reduced tumor blood vessel density, indicating a potential to limit blood and nutrient supply to tumors. Reduced oncogenic miR-21 expression in tumor tissues was observed by in situ hybridization with both curcumin and PLGA–CUR NP treatments. Generated 131I radiolabeled PSMA-monoclonal antibody (mAb) conjugated to PLGA–CUR NPs indicated that the distribution of 131I accumulated at the tumor site dose-dependently (5–20 µL), and greater accumulation occurred with PLGA–CUR NPs conjugated than with 131I radiolabeled PSMA-MAb antibodies alone. Additionally, other organs exhibited minimal 131I PSMA-MAb accumulation, suggesting that PLGA–CUR NP treatment is tumor target specific. These data indicate that PLGA–CUR NPs can significantly accumulate in prostate tissue and exerts prominent anticancer activity [109].

Castrated SCID mice were xenografted with C4-2B cells via intra tibial injection and treated with 1% and 2% curcumin dosing. The first group of animals was treated with curcumin on the same day as intratibial cell injection (preventive model) and the second group was treated 2 months following intratibial cell injection once lesions were established (therapy model). Lesion establishments were found to possess reduced osteosclerotic lesions and slower tumor progression in the curcumin-treated groups compared to the control (0% curcumin diet) [110]. Trabecular bone loss due to tumor growth in the 2% curcumin group was not as significant as in the 0% and 1% groups. Additionally, the 2% curcumin group presented no tumor growth upon curcumin administration the same day as intratibial injections, and PSA levels were also found to be null. The 1% group showed similar results but to a lesser extent. Immunostaining results for osteoblast markers within the bone cell–tumor cell junction were higher in the 0% curcumin group when compared to the 1% and 2% groups, and fatty globules within the injected intratibial area were also more prominent in the 2% group, indicating adipocyte differentiation [110]. TGF-β expression was downregulated in the curcumin groups, whereas bone morphogenic protein (BMP-2) was enhanced, although downstream signaling remained unaffected as shown by the unchanged expression of pSMAD 2/3. Bone morphogenic protein 7 (BMP-7), involved in osteoblast differentiation and EMT, was upregulated in the curcumin-treated groups when compared to the control, and phosphorylation of its downstream effectors SMAD-1, -5, and -8 was also enhanced. BMP-7 overexpression also favored EMT as shown by higher levels of E-cadherin. Higher levels of uncoupling protein-1 (UCP-1) were found in the tumor environment upon curcumin administration, indicative of brown adipose fat (BAT) formation [110].

BALB/c mice subcutaneously injected with PC3 cells were intraperitoneally injected with curcumin for 1 month [93]. A significant reduction in tumor volume by curcumin administration was observed. Curcumin inhibited the mRNA and protein levels of inhibitor of DNA-binding 1 (Id1), which controls tumor aggressiveness. Id1 is a transcription factor involved in several important steps during cancer progression, including cell proliferation, invasiveness, and angiogenesis. These data suggest that the anticancer effects of curcumin may be due to inhibition of Id1 signaling [93].

Treatment of nude mice, subcutaneously injected with PC3 cells, with curcumin resulted in significantly reduced tumor growth, tumor volume, and weight and increased cell apoptosis [111]. In addition, tumor antiapoptotic Bcl-2 protein levels were decreased with curcumin treatment, while proapoptotic Bax protein levels were increased [111].

Treatment of Balb/c nude mice, subcutaneously inoculated with LNCaP cells, with curcumin resulted in significantly reduced tumor growth initially followed by a slow increase in growth over time [112]. The mean prostate-specific antigen (PSA) levels were significantly decreased with curcumin treatment, suggesting reduced prostate cancer tumor presence. In addition, analysis of the tumor tissues revealed that curcumin treatment reduced AR mRNA and protein levels [112].

Combined dietary administration of ursolic acid, curcumin, and resveratrol to FVB/N mice, injected with HMVP2 cell spheroids, resulted in reduced tumor size and weight and increased tumor apoptosis [113]. The combination of ursolic acid, curcumin, and resveratrol resulted in increased glutamine metabolism with significantly reduced glutamine uptake from the extracellular space [113].

BALB/c nude mice, 6 months following subcutaneous injection of human prostate cancer stem cells that were treated with curcumin, had significantly reduced tumor volumes compared to phosphate buffer saline (PBS)-treated control mice [101]. Although the mice injected with curcumin-treated prostate cancer stem cells did eventually develop tumors, the development was delayed, and these tumors had significantly reduced weight and moderate or poor differentiation. Importantly, the mRNA levels of Oct4, Ki67, Psa, and Pap were reduced in these tumors, suggesting reduced cell proliferation. These data suggest that xenografts produced by HuPCaSCs treated with curcumin were smaller and had reduced proliferation capability [101].

The TRAMP (transgenic adenocarcinoma of the mouse prostate) mouse model is a transgenic line of C57Bl/6 mice that develop prostate adenocarcinoma that progresses to metastasis within 30 days. Oral administration of curcumin to TRAMP mice for 30 days significantly reduced testosterone levels in the prostate tissues. Aldo-Keto reductase 1C2 (AKR1C2) protein was localized in nucleus and was also weakly detected in cytoplasm of prostate epithelial cells with curcumin treatment [85]. Curcumin administration had no effect on body weight and prostate volume. These data suggest that curcumin modulates tumor progression by promoting androgen metabolism through AKR1C2 expression [85].

Administration of curcumin, via peritoneal injection, to immunodeficient mice subcutaneously xenografted with LNCap cells, resulted in an inhibition of prostate tumor growth. Analysis of tumor tissues indicated promotion of apoptosis and inhibition of the JNK signaling pathway by curcumin [86]. Curcumin also reduced the level of the epigenetic marker H3K4me3, suggesting a novel epigenetic mechanism for curcumin’s anticancer effects Since curcumin plays a role in regulating epigenetic gene expression, the bromodomain inhibitor JQ-1 was used in combination with it. The inhibition of tumor growth by curcumin and JQ-1 treatment combined was greater than curcumin treatment alone [86].

Intravenous administration of free curcumin, curcumin-loaded liposomes, or aptamer A15 modified (A15)–curcumin-loaded liposomes to mice xenografted with DU145 prostate cancer cells resulted in significantly reduced solid tumor number, tumor size, and weight, while tumor cell apoptosis was increased [114]. A15–curcumin-loaded liposomes had increased curcumin localization within the prostate tumor 12 h after administration (4356 ng×h/tumor weight) compared to free curcumin (1032 ng×h/tumor weight) and curcumin-loaded liposomes (1484 ng×h/tumor weight). These data indicate that the administration of curcumin using A15-modified liposomes improves curcumin’s localization and anticancer properties [114].

Curcumin administration to male Kumming mice, subcutaneously injected with S180 cells, resulted in improved survival and reduced tumor volume and weight [115]. Additionally, curcumin administration increased cell necrosis, lysis, and fragmentation [115].

Administration of curcumin-impregnated poly(lactic-*co*-glycolic) acid (PLGA) to mice subcutaneously injected with prostate cancer cells (PC3, 22rv1, and DU145) resulted in increased apoptosis and fibrosis, as indicated by the increased collagen deposition [116]. Curcumin-impregnated PLGA treatment significantly reduced tumor size, weight, and progression. In addition, Ki67-positive cell levels and monocyte infiltration were reduced with PLGA-impregnated curcumin, suggesting reduced inflammation [116].

Supplementation of *Pten*-deficient mice with a nanoparticle formulation of curcumin (low- or high-dose) had no effect on mouse body, spleen, or prostate weight, while liver weight was significantly increased [117]. Supplementation with high-dose curcumin significantly reduced cancer cell proliferation, and histological assessment of mouse prostate tissue indicated that curcumin reduced the levels of high-grade prostatic intraepithelial neoplasia, increased the incidence of atrophic glands, and increased the thickness of the stroma. These data suggest that daily supplementation of nanoparticle curcumin could prevent early-stage prostate cancer proliferation [117].

## 4. Conclusions

Overall, the available in vitro studies have shown that curcumin is able to inhibit viability, proliferation, survival, migration/invasion, and adhesion of various human prostate cancer cells. Curcumin inhibited both androgen-sensitive and -insensitive prostate cancer cells by targeting a number of signaling cascades responsible for regulating cellular function (Figure 3). The antiproliferative, antisurvival, and antimigratory effects of curcumin in prostate cancer cells may be due to the inhibition of the Akt/mTOR, Ras/MAPK signaling pathways, decreased NF-κB activation, enhanced proapoptoptic caspase and PARP cleavage, and the inhibition of members of the antiapoptotic Bcl-2 family of proteins (Figure 3). Curcumin was also able to induce cell-cycle arrest and enhance autophagy in various prostate cancer cell lines.

The available in vivo studies have shown that curcumin administration is able to inhibit the growth/volume, formation, development, proliferation, and angiogenesis of prostate cancer tumors while promoting apoptosis (Figure 4). These effects were observed in mice xenografted with both androgen-sensitive and -insensitive prostate cancer cells. Curcumin’s inhibition of prostate tumor growth and progression may be due to its inhibition of Akt expression/activation, decreased NF-κB activation, inhibition of the anti-apoptotic proteins Bcl-2 and Bcl-xL, increased expression of the proapoptotic proteins Bax and Bak, and enhanced PARP and caspase expression (Figure 4). These findings from in vivo studies are in agreement with those from the in vitro studies.

The downregulation of cell proliferation, paired with the enhanced activity of programmed cell death both in vitro and in vivo, render curcumin an ideal candidate for the development of novel anticancer pharmaceutical agents providing fewer detrimental effects due to its low toxicity. Future in vitro studies should focus on utilizing cell culture conditions such as different oxygen levels and glucose concentrations for the purpose of obtaining data that represent better the tumor microenvironment seen in vivo. Additionally, further studies utilizing normal prostate epithelium are required to examine whether curcumin can discriminate between cancerous and healthy tissue when interfering with certain signaling pathways. In vivo animal experiments utilizing different prostate cancer models are imperative to accurately determine curcumin dosage and investigate whether curcumin has potent effects against prostate cancer in vivo. Finally, clinical studies are required to examine the effectiveness of curcumin against human prostate cancer.

## Figures and Tables

**Figure 1 biomolecules-10-01536-f001:**
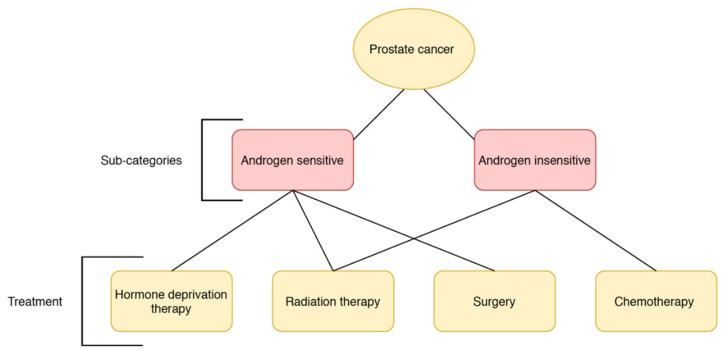
Prostate cancer subtypes and the respective best treatment strategy available.

**Figure 2 biomolecules-10-01536-f002:**
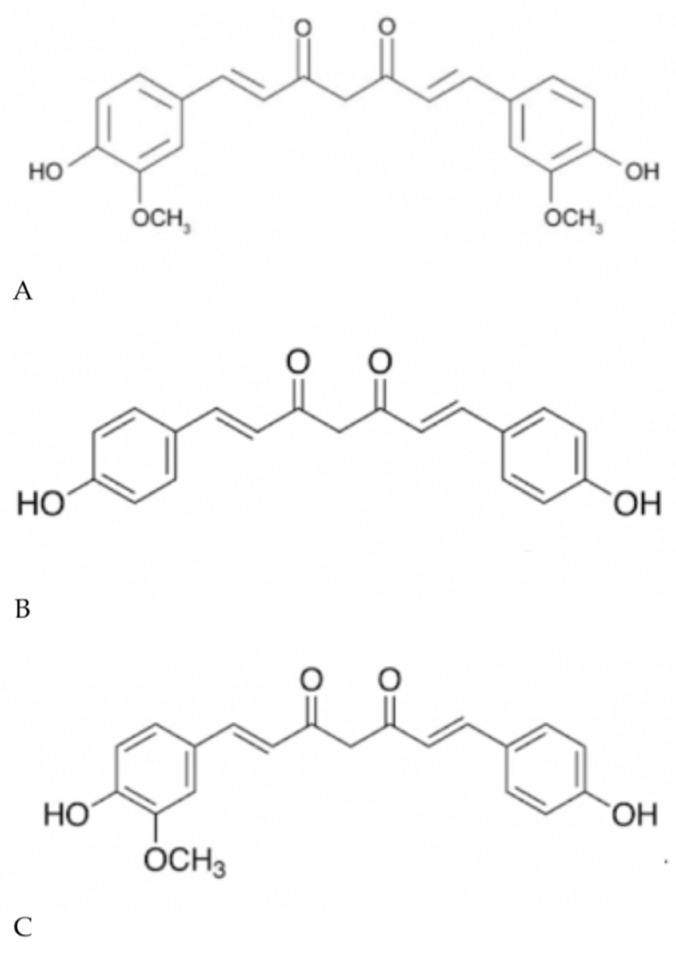
Chemical structure of (**A**) curcumin, (**B**) bisdemethoxycurcumin, and (**C**) demethoxycurcumin.

**Figure 3 biomolecules-10-01536-f003:**
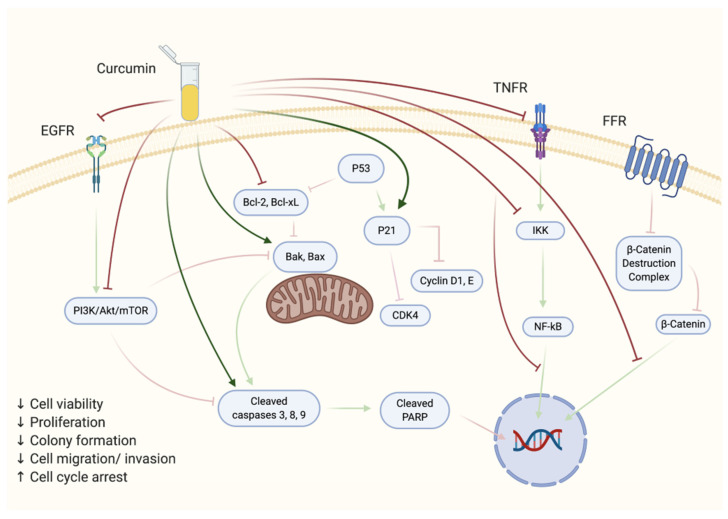
Effects of curcumin treatment on prostate cancer cells in vitro. The figure is based on the data of the studies [54,55,56,57,58,59,60,61,62,63,64,65,66,67,68] and created using BioRender.com.

**Figure 4 biomolecules-10-01536-f004:**
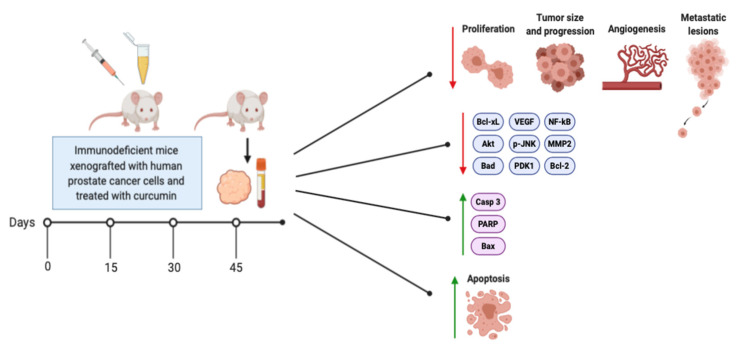
Effects of curcumin administration to animals xenografted with prostate cancer cells. The figure was created based on data from the studies [85,86,88,103,104,105,106,107,108,109,110,111,112,113,114,115,116,117] and created using BioRender.com.

**Table 1 biomolecules-10-01536-t001:** Current available treatments for prostate cancer.

Treatment	Target	Risks and Recurrence
**Prostatectomy**	Efficient against localized and early- or advanced-stage tumors [13]	Moderate biochemical recurrence rate in localized tumors, risk of spreading beyond original location [22]
**Androgen deprivation therapy**	Efficient against androgen- dependent carcinomas	Androgen insensitivity may develop leading to reoccurrence [23]
**Radiation therapy**	Efficient against localized tumors or minimally metastasized tumors [15]	Low efficacy if tumor has developed beyond early stages, requiring combination treatments; higher mortality compared to prostatectomy [24,25]
**Chemotherapy**	Primarily utilized to treat advanced-stage, metastatic, castration-resistant prostate cancer [26]	Improves quality of life and slows disease progression in high-risk prostate cancer, but unlikely to cure it [27,28]

**Table 2 biomolecules-10-01536-t002:** Major prostate cancer cell lines and representative prostate cancer subtypes.

Prostate Cancer Cell Line	Category	Gene Mutations	Representative Prostate Cancer Subtype
**PC3**	Prostatic adenocarcinoma, derived from a bone metastasis of grade IV prostate cancer extracted from a 62 year old Caucasian male [32]	PTEN, p53 [33] PSA-negative Lost AR expression [34]	Androgen-insensitive, highly invasive, rare small-cell prostatic carcinoma [35,36]
**DU-145**	Prostatic adenocarcinoma, derived from a metastasis in the CNS of a 69 year old Caucasian male originating from a primary prostate adenocarcinoma [37]	p53 [38] PSA-negative Lost AR expression [34]	Androgen-insensitive, invasive osteolytic tumor phenotype [36]
**22RV1**	Prostatic adenocarcinoma, derived from a human carcinoma xenograft (CWR22R) serially propagated in nude mice following the castration-induced regression and relapse of androgen dependent CWR22 xenograft [39,40]	p53 [38] AR mutation at codon 874 (His to Tyr) [41] Express endogenous AR PSA-positive [34]	Androgen-sensitive but androgen-independent, low invasiveness, tumors form primarily osteosclerotic lesions [36,42]
**LNCaP**	Prostatic adenocarcinoma, derived from a lymph node metastasis extracted from a 50 year old Caucasian male [43]	PTEN [44] AR gene mutation at codon 868 (Thr to Ala) [45] Express endogenous AR PSA-positive [34]	Androgen-sensitive, localized tumor, fast growth but low invasiveness [35,36,42]
**C4-2B**	Prostatic adenocarcinoma cell lines derived from co-inoculation of LNCaP androgen-dependent cell lines with fibroblasts derived from human osteosarcoma in nude athymic mice for 12 weeks after castration at 8 weeks, and then re-inoculating the extracted tumor with osteosarcoma fibroblasts in castrated mice for another 12 weeks [46]	PTEN [47] PSA-positive Express endogenous AR [34]	Androgen-sensitive but androgen-independent, highly metastatic, primarily form osteoblastic lesions [36,48].

**Table 3 biomolecules-10-01536-t003:** In vitro evidence of the effects of curcumin on androgen-sensitive prostate cancer cells.

Cell Line	Curcumin Dosage	Effects	Reference
LNCaP	0–50 μM; 72 h to assess cell proliferation and cell morphology 20 μM; 24 h to assess expression of Bcl-2, Bcl-xL and Bax 0–50 μM; 24 h to assess PARP cleavage, AR expression, and PSA levels.	↓ Proliferation ↑ Lifted, round cells ↓ Bcl-2 protein ↓ Bcl-xL protein ↑ Phosphatidylserine translocation to outer plasma membrane ↑ PARP cleavage ↓ AR protein ↓ PSA secretion	[54]
LNCaP	10 and 50 μM; 1–4 days to assess cell viability 0–100 μM; 5 h to assess NF-κB expression 50 μM; 0–72 h to assess the expression of Bcl-2 and Bcl-xL 100 μM; 0–72 h to assess the expression and activity of procaspases 3 and 8 50 μM; 0–4 days to assess cell proliferation, PARP cleavage, and apoptosis	↓ Cell viability ↓ Proliferation ↓ NF-κB protein activation ↓ Bcl-2 protein ↓ Bcl- xL protein ↑ Procaspase-3 and -8 activity ↑ PARP cleavage ↑ Apoptosis	[55]
LNCaP	40 μM; 21 days to assess colony formation efficiency 0–50 μM; 24 h to assess cell growth 40 μM; 24 h	↓ Cell growth ↓ AR ↓ AR transcriptional activity ↓ AR transactivation ↓ c-Jun ↓ AP-1 ↓ CBP ↓ Colony formation efficiency	[56]
LNCaP	0–100 μM; 24 h to assess cell proliferation and cyclin D1 protein expression 0–50 μM; 24 h to assess DNA synthesis 25 and 50 μM; 3 and 24 h to assess cell viability 50 μM; 0–180 min and 30–120 min to assess cyclin D1 mRNA expression and CDK4 activity 10 μM; 0–120 min to assess Cyclin D1 promoter activity	↓ Proliferation ↓ DNA synthesis ↓ Cyclin D1 protein and mRNA ↓ CDK4 activity ↓ Cyclin D1 promoter activity	[57]
LNCaP	5–40 μM; 48 h to assess cell viability, caspase activity, DNA fragmentation 10 μM; 20 h to assess cytochrome c release and protein expression of procaspases	↓ Cell viability ↑ TRAIL sensitivity ↑ Cell-cycle arrest at the G2/M phase ↓ Bid protein ↓ Mitochondrial cytochrome c	[58]
LNCaP	35 μM; 0–4 h	↓ p-Akt	[59]
C4-2B	0-15 μM; 1 h to assess EGFR autophosphorylation 1–10 μM; 1 h to assess CSF-1 phosphorylation 7.5 μM and 15 μM; 12 days to assess cell mineralization, IKK activity, and COX-2 expression	↓ EGFR autophosphorylation ↓ CSF-1 phosphorylation ↓ Cell mineralization ↓ IKK activity ↓ COX-2 protein	[48]
LNCaP	10 μM; 48 h to assess cytotoxicity 10 μM; 20 h to assess cleavage of procaspase-3 12.5–50 μM; 20 h to assess NF-κB expression 12.5 μM; 4 and 20h to assess IκBα expression	↑ Cytotoxicity ↑ Apoptosis ↓ NF-κB ↓ P-IκBα ↑ Cleaved caspase-3	[60]
LNCaP	0–40 μM; 24 h	↓ NKX3.1 mRNA and protein ↓ AR mRNA and protein ↓ ARE binding activity	[69]
LNCaP	0–50 μM; 0–72 h	↓ MDM2	[70]
LNCaP	0–30 μM; 48 h to assess cell viability and the expression of Bcl-2, Bax, and Bak 0–30 μM; 3 weeks to assess colony formation efficiency 0–30 μM; 24 h to assess caspase-3 activation and PARP cleavage 0–20 μM; 6–24 h to assess the expression of Bim, Bax, Bak, PMMA, Noxa, Bcl-2, and Bcl-xL genes, and p110, p85, and p-Akt proteins 0–30 μM; 0–24 h to assess mitochondrial membrane potential, release of mitochondrial proteins, translocation of Bax and p53 to the mitochondria, and p53 acetylation and phosphorylation	↓ Cell viability ↓ Colony formation ↑ Caspase-3 activation and nuclear translocation ↑ PARP cleavage ↓ Bcl-2 protein and mRNA ↑ Bax protein and mRNA ↑ Bak protein and mRNA ↑ Bim mRNA ↑ PΜMA mRNA ↑ Noxa mRNA ↓ Bcl-xL mRNA ↓ Mitochondrial membrane potential ↑ Release of Smac/ DIABLO, cytochrome c, and Omi/HtrA2 proteins from mitochondria ↑ Translocation of Bax and p53 to the mitochondria ↑ p53 acetylation and phosphorylation ↑ ROS ↓ p110 ↓ p85 ↓ p-Akt	[61]
LNCaP	10 μM; 48 h to assess cell viability and NF-κB expression 10–30 μM; 24 h to assess Akt phosphorylation 10–30 μM; 24 h to assess protein expression	↓ p-Akt	[71]
LNCaP	25 μM; 17 h	↑ MKP5 mRNA	[72]
LNCaP	0–30 μM; 24 h to assess cell viability 5–40 μM; 21 days to assess colony formation efficiency and TRAIL-induced apoptosis 0–10 μM; 24 h to assess expression of DR4, DR5, DcR1, and DcR2 0–20 μM; 24 and 48 h to assess the expression of pro- and antiapoptotic proteins, caspases, cleaved PARP, and mitochondrial membrane potential	↓ Cell viability ↓ Colony formation efficiency ↑ TRAIL-induced apoptosis ↑ DR4 death receptor ↑ DR5 death receptor ↑ Bak ↑ Bax ↑ PMMA ↑ Bim ↑ Noxa ↓ Bcl-2 ↓ Bcl-xL ↑ Bid cleavage to tBid ↓ IAPs ↓ XIAP ↓ Mitochondria membrane potential ↑ Caspase-3, -8, and -9 cleavage ↑ PARP cleavage	[62]
LNCaP	20 μM; 24 h to assess cell-cycle progression 0–20 μM; 48 h and 20 μM; 0-48 h to assess apoptotic processes 0–20 μM; 0–48 h to assess protein expression	↑ Cell cycle arrest at the G1/S phase ↑ Apoptosis ↑ p27 ↑ p21 ↓ Cyclin D1 ↓ Cyclin E	[63]
LNCaP, C4-2B	0–100 μM; 24–74 h to assess cell proliferation 10 μM; 3–48 h to assess gene expression 5–20 μM; 12 h to assess protein and gene expression, and PSA ELISA	↓ Proliferation ↓ Cell growth ↑ Methionine tRNA synthase ↑ Hemeoxygenase decyclizing ↑ Transcription corepressor activity ↓ Kallikrein 2, 3 ↓ NEDD4- binding protein ↓Transmembrane proteases ↓ Cyclin B1 ↓ AR protein ↓ NKX3.1 ↓ PSA ↓ ERBB2 ↓ EGFR	[73]
LNCaP	0–80 μM; 24–72 h to assess cell proliferation 0–20 μM; 24 h	↓ Proliferation ↓ PSA protein and mRNA ↓ AR mRNA ↓ IL-6	[74]
LNCaP	10 μM, 25 μM, or 50 μM; 24 h	↓ Cell viability ↑ Apoptosis ↓ Necrosis	[75]
TRAMP-C2	0–100 μM; 24 and 72 h	↓ Cell growth ↓ Gli1 (Hedgehog signaling)	[76]
LNCaP	25, 50, and 100 μM; 24–72 h to assess apoptosis and DNA fragmentation	↑ Apoptosis ↑ Ceramide ↑ ssDNA	[66]
LNCaP	0–100 μM; 24 and 48 h to assess cell proliferation 0–30 μM; 24 h to assess AR expression 25 μM; 24 h to assess expression of proteins in the Wnt/β-catenin signaling pathway	↓ Proliferation ↓ AR↓ Nuclear β-catenin ↓ GSK-3b ↓ c-myc ↓ Wnt/β-catenin pathway	[64]
22RV1, LNCaP	10–100 μM; 4 and 24 h 20 μM; 24 h in 22RV1 and DU-145 only for fluorescent microscopy and nuclear staining	↑ Curcumin compartmentalization within cytoplasm and exclusion from nucleus ↑ Cytotoxicity ↑ Apoptosis ↑ Autophagy ↑ LC3B-II isoform ↑ Cell-cycle arrest at the G2 stage ↓ Cyclin B1 ↓ PCNA ↓ β-catenin signaling ↓ c-myc mRNA ↓ Survivin mRNA ↓ Cyclin D1 mRNA ↓ TCF-4 ↓ CREB binding protein ↓ P300	[77]
LNCaP	10 μM; 24 h	↓ AR ↓ PSA	[78]
TRAMP-C1	2.5 and 5 μM; 5 days to assess protein and gene expression 5–100 μM; 1 h to assess methylation	↓ Hypermethylation of CpG sites in Nrf2 ↑ Nrf2 ↑ NQO-1 ↓ CpG methylase (M.Sssl) activity	[79]
LNCaP	5 μM; 7 days	↑ Cytotoxicity ↓ Tri-methylation of H3K27 (H3K27me3) ↓ Methylation of Neurog1 gene ↑ Neurog1 ↑ HDAC1 ↑ HDAC4 ↑ HDAC5 ↑ HDAC8 ↓ HDAC3 ↓ HDAC activity	[80]
LNCaP, 22RV1	20 μM; 24 h 0–50 μM; 24 to assess AR expression	↓ AR ↑ HSP90AA1, GJA1, PRKCE, XRCC6, MIR-141, MIR-183, HMGB1	[81]
LNCaP, C4-2B	5–40 μM; 48 h to assess cell proliferation 20 μM; 24 h for immunoblotting and PCR 15 μM; 1 h for aggregation assay 5 and 10 μM; 24 h for Boyden’s chamber assay	↓ Proliferation ↓ β-catenin signaling ↓ Nuclear β-catenin ↑ β-catenin localization in membrane ↑ Cell–cell aggregation ↑ P-PKD1 ↑ Inactive p-cofilin	[82]
LNCaP, CW22RV1, C-33	0–50 μM; 16 h 25 and 50 μM; 30 min acute treatments	↓ Cell migration/invasion ↓ Total and activated matripase	[83]
LNCaP	25 μM; 48 h to evaluate apoptosis, gene and protein expression 25 μM; 24 h to assess H_2_O_2_ production 1.6–25 μM; 48 h to assess cell viability 10–50 μM; 30 min to assess ROS production	↓ Cell viability ↑ Cytotoxicity ↑ Apoptosis ↓ Bcl-2 ↑ Bax ↑ ROS ↑ CuZnSOD ↑ TRX1 oxidation ↑ TRXR1 mRNA	[84]
LNCaP, 22RV1	0–50 μM; 0–72 h to assess cell proliferation 0–50 μM; 6 h for apoptosis assay 0–50 μM; 24h to assess protein expression and levels of DHT	↓ Cell viability ↑ Caspase-3/7 ↓ DHT ↑ AKR1C2 ↑ SRD5A1 ↓ StAR ↓ CYP11A1 ↓ HSD3B2	[85]
LNCaP	10 U/mL 74 h to assess cell viability	↑ Apoptosis ↓ p-JNK ↓ c-Jun ↓ Bcl-2 mRNA ↓ Tertiary methylation of H3K4	[86]
LNCaP, C4-2B	10 μM; 3–48 h to assess gene expression	↑ Apoptosis ↑ Cell-cycle arrest ↓ Myc ↑ Heme Oxygenase-1 ↑ Cyclic AMP-dependent transcription factor ↓ RAF1 ↓ RAF1 ↓ IGF1R ↓ BCL6 ↑ PTEN ↑ EGFR1 ↑ SMAD ↑ FOXO3 ↑ Akt1 ↑ RAD51 ↓ SOX4 ↓ EGFR ↓ WT1 ↓ E2F2 ↓ MALAT1↑ BMP receptor signaling ↑ PTEN-regulated cell-cycle arrest ↓ TGF-b receptor signaling ↓ WNT signaling ↓ AP-1 ↓ NF-κB signaling ↓ PI3K/Akt/mTOR signaling ↓ FOXM1 ↑ IL-6 signaling ↑ FTH1 ↑ CPEB4 ↑ C6orf61 ↓ PMEPA1	[87]

**Table 4 biomolecules-10-01536-t004:** In vitro evidence of the effects of curcumin on androgen-insensitive prostate cancer cells.

Cell Line	Curcumin Dosage	Effects	Reference
PC3	0–50 μM; 0–72 h IC_50_: 10–20 μM	↓ Proliferation ↑ Lifted, round cells	[54]
DU-145	10 and 50 μM; 1–4 days to assess cell viability 0–100 μM; 5 h to assess NF-κB expression 0-50 μM; 5 h to assess AP-1 expression 50 μM; 0–72 h to assess the expression of Bcl-2 and Bcl-xL 100 μM; 0–72 h to assess the expression and activity of procaspases 3 and 8 50 μM; 0–4 days to assess cell proliferation, PARP cleavage, and apoptosis	↓ Cell viability ↓ Proliferation ↓ NF-κB protein activation ↓ AP-1 protein ↓ Bcl-2 protein ↓ Bcl- xL protein ↑ Procaspase-3 and -8 activity ↑ PARP cleavage ↑ Apoptosis	[55]
PC3	30 μM; 21 days to assess colony formation efficiency 0–50 μM; 24 h to assess cell growth 30 μM; 24 h	↓ Cell growth ↓ Colony formation efficiency ↓ AR transcriptional activity ↓ c-Jun ↓ AP-1 ↓ CBP ↓ NF-κB mRNA	[56]
DU-145	5–40 μM; 48 h	↓ Cell viability ↑ Cell cycle arrest at the G2/M phase	[58]
PC3, DU-145	35 μM; 30 min in PC3 14 μM; 30 min in DU-145	↓ p-Akt in PC3 only	[59]
DU-145	0–125 ug/mL; 24–72 h to assess cell proliferation 10 ug/mL; 0–48 h to assess apoptosis 0–100 ug/mL; 0–48 h to assess MMP-2 an MMP-9 secretion 1–15 ug/mL; 24 h to assess MMP-9 protein expression	↓ Proliferation ↑ Apoptosis ↓ MMP-2 secretion ↓ MMP-9 secretion ↓ MMP-9 protein	[88]
PC3	0–50 μM; 0–72 h to assess the expression of MDM2, p21, Bax, and E2F1 0–30 μM; 24 h and 15 μM; 0–20 h to assess MDM2 mRNA expression 0–30 μM; 24 h to assess ETS2 and p-Akt expression 0–30 μM; 48 h to assess apoptosis, cell viability, and proliferation	↓ Cell viability ↓ Proliferation ↓ MDM2 protein and mRNA ↑ P21 ↑ Bax ↓ E2F1 ↓ Bcl-2 ↓ ETS2 ↓ P-Akt ↑ Apoptosis	[70]
DU-145, PC3	0–30 μM; 48 h to assess cell viability 0–30 μM; 3 weeks to assess colony formation efficiency 0–30 μM; 24 h to assess caspase-3 activation and PARP cleavage	↓ Cell viability ↓ Colony formation efficiency ↑ Caspase-3 activation and nuclear translocation ↑ PARP cleavage	[61]
PC3, DU-145	20 μM in DU-145, 30 μM in PC3; 48 h to assess cell viability and NF-κB expression 20–40 μM in PC3, 10–30 μM in DU-145; 24 h to assess Akt phosphorylation 10–30 μM; 24 h to assess protein expression	↓ NF-κB ↓ IκBα ↓ Bcl-2 ↓ Bcl-xL ↓ XIAP	[71]
PC3, DU-145	25 μM; 17 h 10–50 μM; 17 h to assess IL-6 and IL-8 gene expression 10–25 μM; 17 h to assess NF-κB-luc activity	↑ MKP5 mRNA ↓ TNFα ↓ P-p38 ↓ COX-2 ↓ COX-2 mRNA ↓ IL-6 mRNA ↓ IL-8 mRNA ↓ p38-mediated inflammatory signaling	[72]
PC3	0–30 μM; 24 h to assess cell viability 5–40 μM; 21 days to assess colony formation efficiency and TRAIL-induced apoptosis 0–10 μM; 24 h to assess expression of DR4, DR5, DcR1, an dDcR2 0–20 μM; 24 and 48 h to assess the expression of pro- and antiapoptotic proteins, caspases, cleaved PARP, and mitochondrial membrane potential	↓ Cell viability ↓ Colony formation efficiency ↑ TRAIL-induced apoptosis ↑ DR4 death receptor ↑ DR5 death receptor ↑ Bak ↑ Bax ↑ PMMA ↑ Bim ↑ Noxa ↓ Bcl-2 ↓ Bcl-xL ↑ Bid cleavage to tBid ↓ IAPs ↓ XIAP ↓ Mitochondria membrane potential ↑ Caspase-3 activity ↑ Caspase-3, -8, -9 cleavage ↑ PARP cleavage	[62]
PC3	20 μM; 24 h to assess cell-cycle progression 0–20 μM; 48 h and 20 μM; 0–48 h to assess apoptotic processes 0–20 μM; 0–48 h to assess protein expression	↑ Cell-cycle arrest at the G1/S phase ↑ Apoptosis ↑ p27 ↑ p21 ↑ p16 ↓ Rb hyperphosphorylation ↓ Cyclin D1 ↓ Cyclin E ↓ CDK4	[63]
PC3	0–100 μM; 24 h	↓ Cell proliferation ↓ Glyoxalase 1 activity ↑ Cytotoxicity ↑ Necrosis	[89]
PC3	0–50 μM; 24 h to assess cell viability, 8h to assess protein and DNA synthesis 0–50 μM; 1 h and 40 μM; 0–2 h to assess protein expression 40 μM; 1 h to assess the expression of p-Akt, p-mTOR, and p-S6	↓ Proliferation ↓ p-Akt ↓ p-mTOR ↓ p70S6K ↓ FOXO1 ↓ GSK3β ↑ p-AMPK ↑ MAPK ↓ Cyclin D1 ↑ Phosphatase activity	[65]
PC3	30 μM; 18 h	↓ CCL2 triggered cell adhesion ↓ Cell invasion ↓ Cell motility ↓ Adhesion to fibronectin ↓ CCL2 mRNA ↓ CCL2	[90]
Pc-Bra1	10 μM, 25 μM, or 50 μM; 24 h	↓ Cell viability ↑ Apoptosis ↓ Necrosis	[75]
PC3	25, 50, and 100 μM; 24–72 h to assess apoptosis and DNA fragmentation 100 μM; 3 and 6 h to assess expression of JNK and p38 MAPK 100 μM; 48 h to assess expression of caspases and cytochrome c	↑ Apoptosis ↑ Ceramide ↑ ssDNA ↑ JNK ↑ P38 MAPK ↓ Procaspases-3, -8, and -9 ↑ Accumulation of cytochrome c in cytoplasm	[66]
PC3, DU-145	10–100 μM; 4 and 24 h 20 μM; 24 h in 22RV1 and DU-145 only for fluorescent microscopy and nuclear staining	Curcumin compartmentalization within cytoplasm and exclusion from nucleus ↑ Cytotoxicity ↑ Apoptosis ↑ Autophagy	[77]
PC3	50 μM; 24 h	↓ Proliferation Cell cycle arrest at G2/M phase ↑ Apoptosis ↓ NF-κB ↓ AP-1	[91]
PC3	20 μM; 24 h	↑ IL-6, INS, DDIT3, NDRG1, MIR-152	[81]
PC3	15 μM; 24 h	↓ Iκb kinase β ↑ IκBα ↓ CXCL1 ↓ CXCL2 ↓ NF-κB	[92]
PC3, DU-145,	0–50 μM; 16 h, 24 h 25 and 50 μM; 1 h acute treatments	↓ Cell growth ↓ Cell migration/invasion ↓ Total and activated matripase	[83]
PC3	0–20 μM; 48 h	↓ Cell viability ↑ Apoptosis ↓ ld1 ↓ ld1 mRNA	[93]
PC3, DU-145,	40 μM; 0–24 h to assess expression of ERK1/2, SAPK/JNK 0-60 μM; 24 h to assess expression of p65 and MUC1-C 10-100 μM; 24–72 h to assess cell viability	↓ Cell viability ↑ P-ERK 1/2 ↑ P-SAPK/JNK in PC3 and DU-145 ↓ MUC1-C ↓ NF-κB subunit p65	[94]
PC3	25 μM	↓ EMT ↓ IL-6 ↓ ROS ↓ MAOA/mTOR/HIF-1α ↓ Cell invasion	[95]
DU-145	0–50 μM; 48 h to assess cell proliferation 15 μM; 48 h	↑ Cell death ↓ HGF-induced cell scattering ↓ Wound closure ↓ Cell invasion ↑ E-cadherin ↓ Vimentin ↓ c-Met ↓ Snail mRNA ↓ p-ERK	[96]
DU-145	10–50 μM; 24–72 h to assess cell viability and apoptotic activity 25 μM; 48 h for immunoblotting and PCR 25 μM; 24–72 h to assess cell-cycle progression	↓ Proliferation ↑ Apoptosis ↓ NOTCH1 ↓ Cell survival ↓ Cell growth Cell cycle arrest at the G0/G1 stage ↑ CDK inhibitors ↑ P21 ↑ P27	[97]
PC3, DU-145	0–50 μM; 24h	↑ Apoptosis ↑ Autophagy ↑ Cytotoxicity ↑ TFR1 ↑ IRP1	[98]
DU-145, PC3	0–50 μM; 48 h to assess dose–response relationship 25 μM; 0–48 h to assess time–effect relationship 25 μM; 24 h for scratch assay 10 and 50 μM; 24 h for immunoblotting and PCR	↓ Cell viability ↓ Proliferation ↓ Wound closure ↓ MT1-MMP mRNA ↓ MMP2 mRNA	[99]
DU145, PC3	10 μM; 1–5 days to assess cell viability and migration 10 μM; 0–24 h for PCR and immunoblotting 10 μM; 48 h to assess PGK1 expression	↑ miR-143 ↓ Proliferation ↓ Cell migration ↓ KRAS signaling ↑ Docetaxel sensitivity ↓ PGK1 ↑ FOXD3	[100]
PC3	5 μg/mL; 72 h	↓ Cell viability Cell-cycle arrest ↑ Caspase-3 ↑ Uncleaved caspase-3 ↑ Uncleaved caspase-9 ↑ Caspase-12 ↑ PARP ↑ GRP78 ↑ Inositol-requiring enzyme 1 ↑ Calreticulin ↑ P-eIF2α ↑ Autophagy ↑ LC3B	[67]
PC3	25 μM; 48 h to evaluate apoptosis, gene and protein expression 25 μM; 24 h to assess H_2_O_2_ production 1.6–25 μM; 48 h to assess cell viability 10–50 μM; 30 min to assess ROS production	↓ Cell viability ↑ Cytotoxicity ↑ Apoptosis ↑ ROS ↑ CuZnSOD ↑ TRX1 oxidation ↑ TRXR1 mRNA	[84]
PC3, DU-145	0–20 μM; 4 days to assess cell viability 0–10 μM; 4 days to assess proliferation, protein expression and miR-34a expression	↓ Cell viability ↓ DNA synthesis ↓ Proliferation ↓ Cyclin D1 ↓ PCNA ↑ P21 ↑ miR-34α ↓ β-catenin ↓ c-myc	[68]

**Table 5 biomolecules-10-01536-t005:** In vitro evidence of the effects of curcumin on prostate cancer stem cells.

Cell Line	Curcumin Dosage	Effects	Reference
22RV1 and DU-145 stem cells	46.5 μM; 24 and 48h	↓ Proliferation ↑ miR-145 ↓ lncRNA-ROR ↑ miR-3127 ↑ miR-3178 ↑ miR-1275 ↑ miR-3198 ↑ miR-1908 ↓ Ccnd1 ↓Cdk4 ↓Oct4 ↓CD44 ↓CD133 ↓CCND1	[101]
22RV1 and DU-145 stem cells	46.5 μM; 24 and 48 h to assess proliferation	↓ Proliferation ↓ Cell migration ↑ miR-770-5p ↑ miR-411 ↑ miR-1247 ↓ miR-382 ↓ miR-654-3p	[102]

**Table 6 biomolecules-10-01536-t006:** In vivo evidence of the effects of curcumin on prostate cancer.

Animal Model	Curcumin Dosage	Effects	Reference
Heterotopically implanted LNCaP cell tumors in athymic nude mice	2% composition of a synthetic diet; 6 weeks	↓ Proliferation ↓ Tumor growth ↑ Apoptosis ↑ Pycnotic brown staining nuclei ↓ Mitosis ↓ Nuclear cytoplasm ratio ↑ Fibrotic characteristics ↓ Angiogenesis ↓ Microvessel density ↓ CD31	[103]
SCID mice implanted with DU-145 cells	5 mg/kg b.w./3 times per week; 4 weeks	↓ Tumor volume ↑ Apoptosis ↓ Cancer progression ↑ Caspase-3 activity ↓ MMP-2	[88]
NCr nude male mice injected with PC3 cells	6 µM three times per week; 4 weeks	↑ Apoptosis ↓ Tumor growth ↓ Proliferation ↑ Caspase-3 ↑ PARP ↓ Akt ↓ GSK3Bα ↓ BAD ↓ IKKBα ↓ IκBα	[104]
PC3 tumor-bearing nude mice	5 mg/kg b.w./5 times per week; 4 weeks	↓ Tumor growth ↑ Gemcitabine effect ↑ Irradiation effect ↓ MDM2	[70]
Balb c nude mice implanted with TRAIL-resistant LNCaP cells	30 mg.kg b.w./3 times per week; 6 weeks	↑ Apoptosis ↓ Proliferation ↑ TRAIL-R1/DR4 ↑ TRAIL-R2/DR5 ↑ Bax ↑ Bak ↑ p21/WAF1 ↑ p27/KIP1 ↓ Cyclin D1 ↓ VEGF ↓ uPA ↓ MMP-3 & -9 ↓ Bcl-2 ↓ Bcl-XL ↓ NF-κB activation ↓ Number of blood vessels ↓ Circulating EGFR-2-positive endothelial cells	[105]
TRAMP mice with prostate adenocarcinoma	1–2% dietary composition; 10 or 16 weeks	↓ Tumor formation ↓ Proliferation ↑ Apoptosis ↓ High-grade PIN ↓ Akt ↓ PDK1 ↓ FKHR	[106]
PTEN-KO mice	250 µM; 7 weeks	↓ Prostate adenocarcinomas ↓ Weight gain ↓ Prostate weight ↓ Epithelial cell proliferation ↓ mPIN lesions	[107]
Nude mice injected with PC3 cells	10 µM; 24 h	↓ PC3 cell proliferation ↓ Tumor growth ↓ Tumor progression ↓ VIP mRNA ↓ MMP-2 ↓ MMP-9 ↓ VEGF mRNA and protein ↓ VPAC_1_ receptor density	[108]
CD-1 Foxn1nu male mice injected with PC3 cells	1% diet; 5 weeks	↓ Lung metastasis ↓ Tumor size ↓ Human p53 protein ↓ Proliferating Ki-67 positive cells ~ Tumor morphology	[92]
Athymic nude mice inoculated with C4-3 cells	25 µg; intratumoral injection once	↓ Tumor growth ↑ β-catenin subcellular localization	[82]
Male nude mice inoculated with luciferase-expressed PC3 cells	100 mg/kg b.w.; 3 weeks	↓ Tumor growth ↓ Metastasis ↓ Metastatic lesions ↓ Activated matriptase	[83]
Athymic mice injected with C4-2 cells	25 µg; intratumoral injection once	↓ Tumor volume ↓ Tumor blood vessel density ↑ PLGA–CUR NP curcumin accumulation ↓ Developed vasculature ↓ Bcl-xL ↓ Nuclear AR ↑ β-catenin membrane staining ↓ Nuclear β-catenin activity ↓ CD31β ↓ miR-21	[109]
SCID mice implanted with C4-2B cells	1–2% diet; 4 weeks	↓ Tumor progression ↓ Tumor growth ↓ Osteosclerotic lesions ↓ PSA levels ↑ Osteoblast markers ↑ Fatty globules ↓ TGF-β ↑ BMP-2 ↑ BMP-7 ↑ SMAD-1,-5,-8 ↑ UCP-1	[110]
BALB/c mice injected with PC3 cells	100 mg/kg b.w./day; 1 month	↓ Tumor volume ↓ Id1 mRNA and protein ↑ Id1 cytoplasm locatization	[93]
Balb/c nude mice administered prostate PC3 cells	25 mg/kg b.w./day; 30 days	↑ Apoptosis ↓ PC3 cell growth ↓ Tumor volume and weight ↓ Bcl-2 ↑ Bax	[111]
Balb/c nude mice subcutaneously inoculated with LNCaP cells	500 mg/kg b.w./3 times per week; 4 weeks	↓ Prostate cancer tumor ↓ Tumor growth ↓ PSA level ↓ AR mRNA and protein levels	[112]
FVB/N mice injected with HMVP2 spheroids	1.0% diet; 32 days	↑ Apoptosis ↓ Tumor size ↓ Tumor weight ↑ Glutamine metabolism ↓ Glutamine uptake	[113]
BALB/c nude mice subcutaneously injected with CD44^+^/CD133^+^ HuPCaSCs pretreated with curcumin	IC_50_; 48 h	↓ Tumor size ↓ Tumor development ↓ Oct4 ↓ Ki67 ↓ PSA ↓ Pap	[101]
Immunodeficient mice subcutaneously xenografted with LNCap cells	30 mg/kg; 50 days	↓ Tumor size ↑ Apoptosis ↓ Phospho-JNK ↓ Phospho-c-Jun ↓ Bcl-2 ↓ Bcl-xL ↓ H3K4me3	[86]
Male TRAMP mice	200 mg/kg b.w./day; 1 month	↓ Testosterone level ↓ AKR1C2 surface expression	[85]
DU145 xenograft mice	25 µg; 7 days or 12 h	↓ DU145 cell solid tumors ↓ Tumor size and weight ↑ Tumor localization	[114]
Male Kumming mice injected with S180 cells	18.8 mg/kg b.w./day; 10 days	↑ Survival ↓ Tumor volume ↓ Tumor weight ↑ Necrosis ↑ Cell lysis ↑ Cell fragmentation	[115]
Prostate cancer CD1 mice xenografts (PC3, 22rv1, and DU145 cell-lines)	800 mg/kg; days 1, 9, 18, 27, and 34	↑ Apoptosis ↑ Fibrosis ↓ Inflammation ↓ Tumor size ↓ Tumor weight ↓ Tumor progression ↑ Collagen deposition ↓ Monocyte infiltration ↑ TUNEL-positive cells	[116]
*Pten*-deficient mice	76 or 380 mg/kg b.w./day; 16 weeks	↓ Proliferation ↑ Liver weight ↓ High-grade PIN ↓ Incidence of atrophic glands ↑ Stroma thickness	[117]

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
