# Peer review of "Curcumin against Prostate Cancer: Current Evidence"

_biomolecules, 2020, doi:10.3390/biom10111536_

Round 1

Reviewer 1 Report

The authors reviewed effects of curcumin with in vitro and in vivo studies on prostate cancer. This review contains many results from the related papers, but dose not summarize results well. In addition, the textual content and the content of the tables are duplicated. The authors should summarize the content of the text and remain the tables. There are some minor suggestions as described below.

  • In clinically, castrate-resistant prostate cancer (CRPC) is usually used instead of androgen-insensitive prostate cancer.
  • Authors should change from “apoptosisthat ” to “apoptosis that” on page 11, line 300.
  • Authors should check “H2O2” on page 12, line 377 and in Table 3.

Author Response

Reviewer 1

The authors reviewed effects of curcumin with in vitro and in vivo  studies on prostate cancer. This review contains many results from the related papers, but dose not summarize results well. In addition, the textual content and the content of the tables are duplicated. The authors should summarize the content of the text and remain the tables. There are some minor suggestions as described below.

  • In clinically, castrate-resistant prostate cancer (CRPC) is usually used instead of androgen-insensitive prostate cancer.
  • Authors should change from “apoptosisthat ” to “apoptosis that” on page 11, line 300.
  • Authors should check “H2O2” on page 12, line 377 and in Table 3.

We thank the reviewer for these comments. Following the reviewer’s suggestion, we have reduced the textual content. All minor suggestions have been addressed and corrected within the text. 

Importantly, we have added two figures (Figures 3 & 4) to visually address how curcumin affects prostate cancer both in vitro and in vivo in order to summarize the content of the text. All these changes enhance the quality of our manuscript.

Figure 3: Effects of curcumin treatment on prostate cancer cells in vitro. The figure is based on the data of the studies (54-60, 63, 66, 67, 73, 87, 72, 99, 100) and created using BioRender.com

Figure 4: Effects of curcumin administration to animals xenografted with prostate cancer cells. The figure was created based on data from the studies (82, 83, 85, 103-118). Created by using BioRender.com 

Reviewer 2 Report

About my opinion, curcuminoids such as curcumin have great potential for enchant cancer treatment. This review described numerous example possible curcumin effects for the treatment of prostate cancer. Present topic is suitable for Biomolecules. Nevertheless, some points muss be done before acceptation of manuscript.

Authors presented and discusses curcumin on the various vitro and vivo models. Hover I missing more complex model of curcumin effects in the context tumour biology and biochemistry. Possible way could be based on the schemas, or figures of important signalling pathways in prostate cancers. Also, their importance for prostate cancer should be mentioned and discussed.

Minor

In presented tables should be mentioned, if there is a presented curcumin effects, dose dependently, or not.

Author Response

Reviewer 2

About my opinion, curcuminoids such as curcumin have great potential for enchant cancer treatment. This review described numerous example possible curcumin effects for the treatment of prostate cancer. Present topic is suitable for Biomolecules. Nevertheless, some points muss be done before acceptation of manuscript.

Authors presented and discusses curcumin on the various vitro and vivo models. Hover I missing more complex model of curcumin effects in the context tumour biology and biochemistry. Possible way could be based on the schemas, or figures of important signalling pathways in prostate cancers. Also, their importance for prostate cancer should be mentioned and discussed.

Minor

In presented tables should be mentioned, if there is a presented curcumin effects, dose dependently, or not.

We thank the reviewer for their comments. We have added figures (Figures 3 & 4), into the review that highlight the important signaling pathways and effects of curcumin both in vitro and in vivo.

The dose-dependent effects of curcumin were included in the body of the review, whereas a more comprehensive list of major effects was included within the tables. 

Figure 3: Effects of curcumin treatment on prostate cancer cells in vitro. The figure is based on the data of the studies (54-60, 63, 66, 67, 73, 87, 72, 99, 100) and created using BioRender.com

Figure 4: Effects of curcumin administration to animals xenografted with prostate cancer cells. The figure was created based on data from the studies (82, 83, 85, 103-118). Created by using BioRender.com 

Round 2

Reviewer 2 Report

I have no objection.